# The AquaVIT-4 intercomparison of atmospheric hygrometers

Simone Brunamonti<sup>1</sup>, Harald Saathoff<sup>2</sup>, Albert Hertzog<sup>3</sup>, Glenn Diskin<sup>4</sup>, Masatomo Fujiwara<sup>5</sup>, Karen Rosenlof<sup>6</sup>, Ottmar Möhler<sup>2</sup>, Béla Tuzson<sup>1</sup>, Lukas Emmenegger<sup>1</sup> Nadir Amarouche<sup>7</sup>, Georges Durry<sup>3</sup>, Fabien Frérot<sup>7</sup>, Jean-Christophe Samake<sup>7</sup>, Claire Cenac<sup>3</sup>, Julio Lopez<sup>3</sup>, Paul Monnier<sup>3</sup> and Mélanie Ghysels<sup>8</sup>

<sup>1</sup>Laboratory for Air Pollution / Environmental Technology, Empa, 8600 Dübendorf, Switzerland <sup>2</sup>Institute of Meteorology and Climate Research – Atmospheric Aerosol Research, Karlsruhe Institute of Technology, 76344 Eggenstein-Leopoldshafen, Germany

<sup>3</sup>Laboratoire de Météorologie Dynamique, IPSL, Sorbonne Université, Ecole Polytechnique, 91 128 Palaiseau, France <sup>4</sup>NASA Langley Research Center, Hampton, VA 23681, USA

<sup>5</sup>Faculty of Environmental Earth Science, Hokkaido University, Sapporo, 060-0810, Japan

<sup>6</sup>NOAA Chemical Sciences Laboratory, CSL-8, Boulder, CO 80305, USA

<sup>7</sup> INSU Division Technique, 1 avenue de la terrasse, 91190 Gif-sur-Yvette, France

8 Groupe de Spectrométrie Moléculaire et Atmosphérique, Université de Reims, CNRS, UFR Sciences Exactes et Naturelles, Moulin de la Housse B.P. 1039, 51687 Reims Cedex 2, France

Correspondence to: Mélanie Ghysels (melanie.ghysels-dubois@cnrs.fr)

**Abstract**. The AquaVIT-4 intercomparison of atmospheric hygrometers was conducted at the AIDA climate simulation chamber of the Karlsruhe Institute of Technology (KIT), Germany, in March-April 2022, within the framework of the HEMERA H2020 EU project. The objectives were to document the performance of existing hygrometers and to support the development of novel methods for water vapor (H<sub>2</sub>O) measurements in the upper atmosphere. The AquaVIT-4 intercomparison involved seven hygrometers, based on either infrared laser absorption spectroscopy or frostpoint hygrometry techniques: four deployed on aircraft or stratospheric balloon platforms, and three reference instruments. The simulated conditions in the AIDA chamber reproduced the characteristic atmospheric conditions of the upper troposphere-lower stratosphere (UTLS, altitude range ~5– 28 km) in the tropics and mid-latitudes, spanning between 20–600 hPa pressure, 190–245 K temperature, and 0.5–530 ppm H<sub>2</sub>O mixing ratio. The campaign was divided into two phases, each consisting of four measurement days: an "open intercomparison", where the simulated conditions were known to the participants, and a "blind intercomparison", where the conditions were coordinated by independent referees and unknown to the participating teams. Here we present a statistical analysis of the entire dataset, which allows to assess the accuracy and limitations of each instrument. For the accuracy evaluation, two sets of reference measurements were defined: one for in situ instruments, located inside the AIDA vessel, and one for extractive instruments, sampling the chamber gas through a heated inlet. This distinction accounts for H<sub>2</sub>O desorption effects, which are most prominent at low pressures and low H<sub>2</sub>O concentrations. All instruments showed a good agreement with the reference values in the range of  $H_2O > 2$  ppm, with mean deviations within  $\pm 7$  % for  $H_2O > 10$  ppm, and  $\pm 8$  % between 2–10 ppm  $H_2O$ . The largest differences were found for  $H_2O < 2$  ppm, a rarely observed range in the atmosphere, though most of the instruments still achieved average deviations within ±10 %. Overall, the results of AquaVIT-4 demonstrate the high accuracy and reliability of the four involved sensors for upper atmospheric monitoring and research applications.

# 1. Introduction

15

20

Water vapor (H<sub>2</sub>O) is a strong greenhouse gas that substantially contributes to the Earth's radiative balance. Radiative-convective models and observations have shown that the increasing greenhouse gases like CO<sub>2</sub> can lead to a moistening of the troposphere (Soden et al., 2005; Dessler et al., 2008; 2013), potentially doubling the warming effect of CO<sub>2</sub> (Banerjee et al., 2019; Dessler et al., 2013). Dessler et al. (2013) demonstrated a link between interannual variations in tropospheric temperature and the amount of H<sub>2</sub>O entering the stratosphere, suggesting a stratospheric water vapor climate feedback of about +0.3 W m<sup>-2</sup> K<sup>-1</sup> (see also Forster et al., 2021 for an overview of the topic). Observational studies showed that a stratospheric moistening can lead to an increase in the mean surface temperature (Forster and Shine, 1999; Riese et al., 2012; Solomon et al., 2010), with disparities across different latitudes. Additionally, stratospheric H<sub>2</sub>O affects the atmospheric circulation (Maycock et al., 2014) and stratospheric ozone chemistry (Stenke and Grewe, 2005).

In situ measurements by the NOAA frost-point hygrometer (FPH) reported a 1–1.5 % annual increase in stratospheric H<sub>2</sub>O (16–28 km altitude) between 1980 and 2000 (Oltmans et al., 2000). Using multiple datasets, Rosenlof et al. (2001) found a persistent trend of 1 % annual increase in stratospheric H<sub>2</sub>O from 1954 to 2000, later revised to 0.6 % per year (Scherer et al., 2008). Statistically significant positive trends of stratospheric H<sub>2</sub>O were also found by Hurst et al. (2011; 2022), along with short-term variability driven by cold-point tropopause (CPT) temperature variations (e.g., Randel and Park, 2019) and large-scale stratospheric dynamics, such as the Brewer-Dobson circulation (Tao et al., 2023) and the quasi-biennial oscillation (Tian et al., 2019). The analysis of a composite of satellite observations showed negative trends in H<sub>2</sub>O in the lower and mid-stratosphere, and positive trends in the upper stratosphere, due to methane oxidation (Hegglin et al., 2014; Tao et al., 2023).

In the upper troposphere and lower stratosphere (UTLS, altitude range ~5–28 km), and in particular above the tropopause, typical mixing ratios of H<sub>2</sub>O are < 10 ppm (parts per million, i.e. μmol/mol). High-resolution measurements above 15 km are primarily obtained from instruments carried on stratospheric balloons or aboard high-altitude aircraft, while satellite observations provide vertical profiling with 1.5–3 km resolution and extensive spatial and temporal coverage (e.g., Hegglin et al., 2013; Hurst et al., 2014; 2016). Although this allows to address large-scale processes, the use of in situ instruments aboard balloons or aircraft remain essential for resolving microphysical processes and to validate satellite observations. In particular, high-resolution observations are crucial for studying H<sub>2</sub>O transport and dehydration processes in the UTLS, such as fine cirrus layer formation near the tropopause (e.g., Luo et al., 2003; Peter et al., 2006). Numerical Weather Prediction (NWP) models largely benefit from water vapor observations in the UTLS in validating their models as stratospheric H<sub>2</sub>O is not assimilated. Discrepancies up to ±17 % have been found between operational and re-analysis (ERA5) data of the European Centre for Medium-Range Weather Forecasts (ECMWF) and balloon-borne measurements (Brunamonti et al., 2019).

Upper air H<sub>2</sub>O is classified as an essential climate variable (ECV) by the Global Climate Observing System (GCOS), with revised measurement requirements (GCOS, 2022). Particularly, the minimum threshold for useful measurements in terms of climate monitoring studies is set to 0.5 ppm (~10 %) at 250 m vertical resolution, whereas the "goal" level, for which no further improvement is necessary, is set to 0.1 ppm (~2 %) at 10 m resolution. Currently, routine measurements of UTLS H<sub>2</sub>O in the

GCOS Reference Upper Air Network (GRUAN) are performed by balloon-borne cryogenic frost-point hygrometry (CFH/FPH). These instruments are based on the chilled mirror principle and have an estimated uncertainty of 4–6 % in the UTLS (Hall et al., 2016; Vömel et al., 2016). However, these devices are being redesigned due to the phase-out of the cooling agent fluoroform (HFC-23) used for their operation because of its high global warming potential (UNEP, 2016). Thus, there is an urgent need for alternative, reliable technologies for the long term UTLS H<sub>2</sub>O monitoring. Alternative cooling solutions, such as the use of liquid nitrogen or a mix of dry ice and alcohol, are currently being implemented and validated (e.g., Rolf et al., 2020; Dirksen, 2024; Poltera et al., 2025).

Despite their importance, accurate in situ measurements of UTLS  $H_2O$  remain highly challenging, with substantial discrepancies between different techniques. Comparison between Lyman- $\alpha$  fluorescence (e.g., Zöger et al., 1999; Sitnikov et al., 2007) and CFH measurements showed differences within  $\pm 10$  % between 11-20 km altitude, and up to 30 % outside this range (Vömel et al., 2007). Rollins et al. (2014) found differences as large as 20 % between airborne Lyman- $\alpha$ , laser spectroscopy and mass spectrometry instruments, and balloon-based CFH measurements during the MACPEX campaign. Kaufmann et al. (2018) reported deviations as large as 20 % for  $H_2O < 10$  ppm between Lyman- $\alpha$  and mass spectrometry instruments on-board the HALO aircraft. Ghysels et al. (2016) compared the Pico-SDLA  $H_2O$  spectrometer with Lyman- $\alpha$  in the tropical UTLS (15–23 km altitude), finding differences up to  $1.9 \pm 9.0$  % above the CPT. Singer et al. (2022) reported good agreement between off-axis integrated cavity output spectrometry and Lyman- $\alpha$  instruments onboard the Geophysica M-55 aircraft, and balloon-borne CFH measurements during the StratoClim airborne campaign in the Asian UTLS. A recent study by Ghysels et al. (2024) compared the Pico-Light  $H_2O$  hygrometer with the NOAA FPH in the mid-latitudes and found an agreement of  $4.2 \pm 2.7$  % in the UTLS.

10

Considering the above findings, rigorous intercomparisons are of critical importance to allow a valuable scientific interpretation and quality assessment of the measured data (e.g., Krämer et al., 2009). Over the past two decades, this led to the organization of a series of international hygrometer intercomparisons, known as the AquaVIT, held at the AIDA (Aerosol Interaction and Dynamics in the Atmosphere) climate simulation chamber of the Karlsruhe Institute of Technology (KIT), Germany, and aimed to clarify the uncertainties in UTLS  $H_2O$  measurements and to identify the causes of the observed discrepancies. The first AquaVIT-1 intercomparison (Fahey et al., 2014) was conducted in October 2007 at the AIDA chamber, testing more than 20 different water vapor sensors subject to a wide range of environmental conditions (pressure 50-500 hPa, temperature 185-243 K,  $H_2O$  mixing ratio 0.3-152 ppm). Given the absence of an accepted reference measurement, the accuracy of the instruments was established with respect to the ensemble mean of a "core" subset of instruments. While many instruments were found to differ from such a reference by  $\pm 100$  % and more, the "core" instruments showed variations smaller than  $\pm 10$  % relative to the reference value under all conditions (Fahey et al., 2014). The AquaVIT-2 intercomparison was conducted in April 2013 and the results have been reported individually by some of the participating instruments (Meyer et al., 2015; Hall et al., 2016). AquaVIT-3 was conducted in July 2015 with aircraft instruments and new onboard calibration units. The AquaVIT series assessed the quality of major atmospheric hygrometers and documented their improvements over time, facilitated by information exchange during the intercomparison campaigns.

As a continuation of these efforts, the AquaVIT-4 campaign was conceived and conducted in March-April 2022 at the AIDA chamber, within the HEMERA H2020 EU project. The intercomparison featured in total seven hygrometers, based on either direct infrared laser absorption spectroscopy or frostpoint hygrometry techniques: four designed to be deployed on aircraft or stratospheric balloon platforms, and three reference instruments. The simulated conditions in the AIDA chamber were focused on reproducing the characteristic atmospheric conditions of the tropical and mid-latitude UTLS (pressure 20–600 hPa, temperature 190–245 K, H<sub>2</sub>O mixing ratio 0.5–530 ppm).

Here we present the results of the AquaVIT-4 measurement campaign and assess the accuracy and limitations of the participating instruments. The AIDA chamber facility and the individual instruments are described in Sections 2-3. The strategy of the intercomparison and the simulated conditions are presented in Section 4. The statistical analysis of the results is discussed in Sections 5-6. Detailed descriptions of the data processing algorithms and instrument-specific issues are provided in the Appendix.

#### 2. AIDA chamber

The Aerosol Interaction and Dynamics in the Atmosphere (AIDA) chamber, located at Karlsruhe Institute of Technology (KIT, Germany), is an aluminium vessel of 84 m³ volume, with the possibility to control pressure from 1100 hPa to 0.01 hPa, and temperature from 313 K to 183 K (Möhler et al., 2003; Wagner et al., 2009; Skrotzki et al., 2013). A mixing fan inside the AIDA chamber provides homogeneity throughout the chamber volume within 90 s. This allows to simulate atmospheric conditions relevant for aerosol and cloud formation processes under tropospheric and lower stratospheric conditions on minutes to hours timescales. Over the past two decades, the AIDA chamber was used in a large number of studies investigating atmospheric aerosols and clouds (Wagner et al., 2006, Möhler et al., 2008, Donahue et al., 2012, Skrotzki et al., 2013, Lamb et al., 2017, Gao et al., 2022), as well as to test and compare the performance of research instruments to be deployed in field campaigns (Laborde et al., 2012, Shen et al., 2024), including atmospheric hygrometers. Most notably, the AIDA chamber hosted the previous three intercomparison campaigns of the AquaVIT series, including AquaVIT-1 (Fahey et al., 2014), involving more than 20 different atmospheric hygrometers and AquaVIT-2 with a focus on traceability to national standards (Buchholz et al., 2014, Hall et al., 2016) as well as AquaVIT-3 testing aircraft instruments and onboard calibration units. The work presented here heavily relies on the AquaVIT-1 experience for the experimental design and integration of the different sensors into the AIDA chamber facility.

The key features of the AIDA chamber for the AquaVIT intercomparisons include: (i) operation at near-constant pressure ( $\pm 1$  hPa) and temperature ( $\pm 0.3$  K) conditions; (ii) possibility to vary the H<sub>2</sub>O mixing ratio, concentration and relative humidity by addition of H<sub>2</sub>O or dry air, and/or the partial removal of chamber air by pumping; (iii) large chamber volume with small surface-wall-to-volume ratio, allowing multiple instruments to be located inside the chamber or to sample air from outside the chamber without significantly disturbing internal conditions (Fahey et al., 2014). Conditions (i-ii) allow the H<sub>2</sub>O mixing ratio in the chamber to be constant or slowly changing for long periods of time (up to  $\sim 1$  hour), thereby allowing adequate time for

all instruments to sample chamber air and make multiple determinations of water vapor content. Condition (*iii*) facilitates the use of customized, extractive sampling probes to deliver chamber air to instruments located outside the chamber. As in Aqua-VIT-1 (Fahey et al., 2014), the probes used here consisted of electropolished stainless-steel tubing and were heated to reduce water adsorption and to evaporate droplets.

# 5 3. Instruments and configuration

30

The AquaVIT-4 intercomparison featured in total 7 atmospheric hygrometers (4 participants and 3 reference instruments), based on either direct infrared laser absorption spectroscopy or frostpoint hygrometry, which were tested in the AIDA chamber under a wide range of pressures, temperatures and humidities, simulating the characteristic atmospheric conditions for an altitude range of approximately 5–28 km, from the mid- to tropical latitudes.

The participating instruments included (in alphabetic order): the balloon-borne laser absorption spectrometer for UTLS water research ("ALBATROSS") by Empa, Switzerland (Graf et al., 2021; Brunamonti et al., 2023); the Diode Laser Hygrometer (DLH) by NASA, USA (Diskin et al., 2002); the Pico-Light H<sub>2</sub>O (Ghysels et al., 2024) by CNRS, France, successor of the former Pico-SDLA instrument (Durry et al., 2008); and the Surface Acoustic Wave frostPoint Hygrometer (SAWfPHY) by LMD, France. Reference instruments, all operated by KIT (Germany), included the AIDA-PCI-in-cloud-TDL (APicT) (Ebert et al., 2005), the single-path AIDA-PCI-in-cloud-TDL (SP-APicT) (Skrotzki, 2012, Sarkozy et al., 2020, Lamb et al., 2023), and the commercial chilled-mirror hygrometer MBW373LX by MBW Calibration AG (Switzerland). The main features and reported uncertainties of all instruments are summarized in Table 1.

Figure 1 illustrates the experimental setup and the installation location of the different instruments in the AIDA chamber. The instruments participating in AquaVIT-4 can be divided in two categories: internal (in situ) instruments, located either directly inside the AIDA vessel or within its thermostated enclosure, and external (extractive) instruments, probing the chamber air through a heated sampling line. Specifically, five instruments participating in AquaVIT-4 are classified as internal (Pico-Light H<sub>2</sub>O, SAWfPHY, DLH, APicT, SP-APicT), and two as external (ALBATROSS, MBW373LX). The Pico-Light H<sub>2</sub>O and SAWfPHY hygrometers were installed inside the main vessel of AIDA, separated by about one meter distance. The main optical components of APicT, SP-APicT and DLH were located within the thermostated enclosure (outside the main vessel), while their optical beams were folded between the inner chamber walls, providing a measurement of the H<sub>2</sub>O mixing ratio averaged over the full diameter of the chamber. Conversely, ALBATROSS and MBW373LX were located outside of the thermostated enclosure, sharing the same heated sampling line. The flow rate drawn by the two instruments varied between 0.3–1 standard liter per minute (SLM) for MBW373LX, and 0.02–0.5 SLM for ALBATROSS, depending on AIDA pressure (see Appendix A3). This corresponds to a total removal of up to 0.002 % of the chamber volume per minute (~100 times smaller than in AquaVIT-1, Fahey et al., 2014).

| Instrument                     | Institute            | Technique                                                      | Reported uncertainty                                                                                                                                                             | Sampling   | Status      |
|--------------------------------|----------------------|----------------------------------------------------------------|----------------------------------------------------------------------------------------------------------------------------------------------------------------------------------|------------|-------------|
| APicT                          | KIT, Germany         | Tunable-diode laser<br>absorption spectros-<br>copy (TDLAS)    | Accuracy $< 5 \%$ . 1- $\sigma$ precision (2 s): ~25 ppb (from ~10 % at 0.25 ppm H <sub>2</sub> O to 0.1–1 % at $> 2$ ppm H <sub>2</sub> O), dynamic range 1–2000 ppm            | In situ    | Reference   |
| SP-APicT                       | KIT, Germany         | Tunable-diode laser<br>absorption spectros-<br>copy (TDLAS)    | Accuracy < 5 % for >100 ppm H <sub>2</sub> O, dynamic range: 100–13000 ppm                                                                                                       | In situ    | Reference   |
| MBW373LX                       | KIT, Germany         | Dewpoint mirror hygrometry                                     | Accuracy $\pm 0.1$ K on frostpoint temperature ( $T_{fp}$ ) for range $T_{fp} > -95$ °C, corresponding to < 2 % in H <sub>2</sub> O mixing ratio                                 | Extractive | Reference   |
| ALBATROSS                      | Empa,<br>Switzerland | Quantum-cascade<br>laser absorption<br>spectroscopy<br>(QCLAS) | Accuracy < 1.5 % for range 2.5–180 ppm $H_2O$ , $p = 30–250$ hPa (*), 1- $\sigma$ precision (1 s) ~30 ppb (~0.1 % at > 30 ppm $H_2O$ )                                           | Extractive | Participant |
| DLH                            | NASA, USA            | Tunable-diode laser<br>absorption spectros-<br>copy (TDLAS)    | Accuracy is the greater of 5 % or 0.5 ppm; Precision (1-σ, 1 s) is greater of 0.1 % or 50 ppbv                                                                                   | In situ    | Participant |
| Pico-Light<br>H <sub>2</sub> O | CNRS, France         | Tunable-diode laser<br>absorption spectros-<br>copy (TDLAS)    | Total uncertainty 12 % for $p < 45$ hPa, 7.5 % for $p = 45$ –70 hPa, 5.5 % for $p = 70$ –85 hPa, ~4 % for $p = 85$ –180 hPa, ~1.5 % for $p > 180$ hPa. Precision (1 s): 130 ppb. | In situ    | Participant |
| SAWfPHY                        | LMD, France          | Surface acoustic wave frostpoint hygrometry                    | Accuracy $\pm 0.1$ K on frostpoint temperature ( $T_{fp}$ ) for range $T_{fp} > -95$ °C, corresponding to $< 2$ % in H <sub>2</sub> O mixing ratio                               | In situ    | Participant |

Table 1. Summary and main features of all instruments participating in the AquaVIT-4 intercomparison. (\*) During AquaVIT-4, additional uncertainty (up to  $\pm 0.2$  ppm  $H_2O$  at low  $p_{AIDA}$ ) due to the extractive configuration (see Appendix A3).

Figure 1. Schematic of the experimental setup used within the AquaVIT-4 intercomparison campaign.

# 3.1. ALBATROSS laser spectrometer

The balloon-borne laser absorption spectrometer for UTLS water research ("ALBATROSS") is a compact (< 3.5 kg) mid-IR laser spectrometer developed by Graf et al. (2021). ALBATROSS incorporates a monolithic segmented circular multipass cell (Graf et al., 2018) that allows an optical path length of 6 m within a cell diameter of 10.8 cm. The multipass cell is highly resistant to thermally induced distortion and it can be operated in both closed and open-path configuration. A continuous wave distributed-feedback mid-IR quantum-cascade laser (cw-DFB-QCL), tuned to a spectral window of ~1 cm<sup>-1</sup> centred around an isolated absorption line of  $H_2O$  at  $1662.809 \text{ cm}^{-1}$  ( $\lambda \approx 6.01 \text{ }\mu\text{m}$ ), is used as a light source. ALBATROSS uses rapid spectral sweeping of the QCL by periodic modulation of the laser driving current. A highly energy-efficient strategy, referred to as "intermittent continuous-wave" (iCW) modulation (Fischer et al., 2014), is implemented, in which the driving current is applied in pulses, typically 200  $\mu$ s long, followed by a short period of complete shutdown of the laser. The transmission data, consisting of 25 × 10<sup>3</sup> data points, are digitized by a 14-bit analogue-digital converter (ADC) at 125 MSs<sup>-1</sup> and real-time processed by an FPGA (STEMlab 125-14, Red Pitaya). The signal-to-noise-ratio is further improved by averaging up to 3000 individual spectra in real time, leading to an effective measurement rate of 1 Hz. The raw spectra are acquired at 3 kHz and eo-averaged to an output resolution of 1 s.- A full description of the laser driving and data acquisition systems can be found in Graf et al. (2021).

The accuracy and precision of ALBATROSS at UTLS-relevant conditions were validated using SI-traceable reference mixtures, generated by a dynamic-gravimetric permeation method established at the Swiss Metrological Institute, METAS (Bruna-

monti et al., 2023). It was demonstrated that ALBATROSS achieves an accuracy better than  $\pm 1.5$  % at all investigated pressures (30–250 hPa) and H<sub>2</sub>O mixing ratios (2.5–35 ppm), and a 1 s precision better than 30 ppb (i.e., 0.1 % at 35 ppm H<sub>2</sub>O). The spectrometer also achieves a linear response within  $\pm 1.5$  % up to 180 ppm H<sub>2</sub>O (Brunamonti et al., 2023).

Recently, ALBATROSS was deployed in a series of atmospheric test flights conducted from the MeteoSwiss Payerne Observatory (Switzerland) in the framework of the Swiss  $H_2O$ -Hub project, in tandem with a CFH instrument as a reference. The results show a good agreement within  $\pm 10$  % between ALBATROSS and CFH up to about 30 hPa pressure (~24 km altitude). For the AquaVIT-4 intercomparison, ALBATROSS was operated in a "closed-path" configuration, as described in Brunamonti et al. (2023), in which the sample gas is pumped from the AIDA chamber through a heated sampling line. A detailed description of the sampling system and the data processing algorithm is given in Appendix A3.

# 3.2. DLH Diode Laser Hygrometer

10

The NASA Langley Diode Laser Hygrometer (DLH) is an external, open-path near-infrared tunable diode laser absorption instrument which has flown on nine different aircraft since 1994 over the past 25+ years (Diskin, et al., 2002). There are several versions of DLH, as they are designed and constructed specifically for each aircraft; in general, each consists of a transmitter/receiver and a quasi-retroreflector, between which is the two-pass optical absorption path. Path lengths vary between approximately 1–30 m, depending on the aircraft. The DLH instrument that was used during AquaVIT-4 (DLH-WB) is the one which has flown on the NASA WB-57F aircraft during several campaigns since 2008, reaching approximately 20 km altitude in flights lasting up to 6 hours (e.g., Rollins et al., 2014).

The transceiver in the DLH-WB is mounted inside an airfoil-shaped fin which is mounted beneath the aircraft wing, and the retroreflective film is typically located on either a thin antenna-like fin mounted beneath an instrument pod, or on an engine nacelle, on the same wing as the transceiver. Optical paths on this aircraft have ranged from approximately 13 m to 18 m. For AquaVIT-4, the DLH-WB instrument housing (which contains the laser itself and all data acquisition and control, as well as the transceiver) was mounted outside the chamber in the cold zone, and the optical beam was passed through a custom-antireflection-coated plane parallel window into and out of the chamber. The retroreflective film was mounted on the inside of a window blank on the opposite side of the chamber, resulting in a two-pass optical path in the chamber of about 9.41 m. The short path between the instrument transceiver window and the facility window was covered by a custom housing and purged continuously with synthetic ultra-dry air. External heaters were added to the instrument housing to maintain a minimum temperature of approximately -20 °C while the instrument was unpowered during the overnight hours. During testing hours, those heaters remained on, but internal temperatures were maintained via the same heaters and control system which operate during flight.

To measure water vapor, the DLH laser is tuned to one of three spectral absorption lines near 1.395 μm. These three lines have line-strengths which each differ by nearly an order of magnitude, allowing the instrument to measure water vapor in the atmosphere over its entire dynamic range. In flight, the DLH control software selects the appropriate absorption line for the

conditions, and switches automatically between them to maintain good instrument performance. During AquaVIT-4, due to the unusual circumstances of the facility operation, operating modes were adjusted manually, and for all reported data, only the strongest line was used.

DLH uses line-locked, multi-harmonic wavelength modulation spectroscopy to achieve high accuracy, high precision and fast temporal response. For AquaVIT-4, only the second harmonic was used to convert signal to water vapor mixing ratio. Because of the dependence of absorption signal on temperature and pressure, both in flight and in ground-based facilities, conversion of DLH data to water vapor mixing ratio requires additional information on those parameters as well as the installed optical path length. For AquaVIT-4, the temperature and pressure data were provided by the AIDA facility operators through a network interface.

# 10 3.3. Pico-Light H<sub>2</sub>O tunable diode laser hygrometer

15

Pico-Light H<sub>2</sub>O is a mid-infrared lightweight tunable diode laser hygrometer, probing water vapor at 2.63 µm. The Pico-Light instrument has primarily been developed for sounding of the upper troposphere and stratosphere. It relies on mid-infrared direct absorption spectroscopy. The beam of a 2.63 µm antimonide laser diode is propagated in the open atmosphere over a 1 m distance; absorption spectra are thereby recorded in situ at 10 ms intervals by ramping the driving laser current. The water vapor mixing ratios are retrieved from the in situ absorption spectra using a molecular model (here Voigt profile) in conjunction with in situ atmospheric pressure and temperature measurements. The absolute pressure is measured by an absolute pressure transducer (precision 0.05 % full scale, absolute uncertainty 0.5 hPa; model PPT1, Honeywell, USA) and the air temperature is measured using two fast response temperature sensors (Sippican, USA) with an uncertainty of 0.2 K root mean square and a resolution of 0.1 K. Their uncertainty was improved by an intercomparison programme with the World Meteorological Organization (WMO) (Nash et al., 2011). One sensor is located at each end of the optical cell. The enhanced electronics are both smaller and more energy efficient compared with those employed in previous Pico-SDLA. This decrease in power consumption and shorter flight duration have contributed to a significant one-third reduction in the energy budget, now standing at just 3.5 Wh.

Pico-Light H<sub>2</sub>O It-was compared in flight with the NOAA FPH (Ghysels et al., 2024) during the AsA 2022 balloon campaign from the CNES Aire-sur-l'Adour (France) balloon launch facility (Ghysels et al., 2024). Each of the instruments were flown under their own free release balloon. Between the mid-latitude lapse rate tropopause and 20 km, the mean relative difference between water vapor mixing ratio measurements by Pico Light H<sub>2</sub>O and NOAA FPH was 4.2 ± 2.7 %. In the middle troposphere, the relative difference was 3.8 ± 23.6 %, with differences depending on the altitude range considered. In the troposphere, relative humidity over water (RH) comparisons leads to an agreement between both Pico Light H<sub>2</sub>O and NOAA FPH of =0.2 % on average, with excursions of about 30 % RH due to moisture variability. The Pico Light H<sub>2</sub>O was also compared to iMet 4 (InterMet, USA) and M20 (Modem, France) sondes in the lower troposphere. Pico Light H<sub>2</sub>O agrees well with iMet-

4 between ground and 7.5 km (within ± 3 % RH) and so does for M20 sondes, up to 13 km, where M20 sondes are wet biased by 3 % RH and dry biased by 20 % in case of saturation.

#### 3.4. SAWfPHY acoustic wave frost point hygrometer

SAWfPHY is a recently developed low-power (3W), lightweight instrument (3 kg) aimed at performing water vapor measurements on-board long-duration balloon flights in the lower stratosphere. SAWfPHY has already performed several month-long flights in the tropical lower stratosphere, in the frame of the Strateole-2 project. SAWfPHY is a frost-point hygrometer. Yet, instead of standard frost-point hygrometers that use optical methods to detect frost on a mirror, the sensing surface of SAW-fPHY is a Surface-Acoustic Wave (SAW) resonator. The sensor, which is cooled by a Peltier thermoelectric device, is enclosed in a stainless-steel chamber, in which air is pumped from the outside. The resonator properties (peak frequency and factor of quality) are modified by the ice deposit on the substrate. The control loop of the instrument is thus designed to maintain a constant and detectable amount of ice on the sensor, at which point the resonator temperature is equal to the frost point temperature. The resonator temperature is measured by a bead thermistor located close to it, and the saturation vapor pressure over ice (Murphy and Koop, 2005) is used to convert the measured frost-point temperature into water vapor partial pressure (and mixing ratio).

During the AquaQUAVIT-4 campaign, SAWfPHY's sensing chamber was located in the main AIDA vessel, next to the Pico-Light H<sub>2</sub>O instrument. On the other hand, the instrument electronics was located outside of the vessel, in the laboratory. The air flow inside the sensing chamber was ensured during AquaVIT-4 by connecting the chamber to the vacuum line of AIDA through a mass flow controller. During balloon flights, the air flow is ensured by a pump working at atmospheric pressure. The instrument performed nominally during the two campaign weeks, except for two days corresponding to the coldest AIDA temperature set points (day 4 of the open intercomparison with  $T_{AIDA} \sim 194$  K, and day 3 of the blind intercomparison with  $T_{AIDA} \sim 190$  K). During those periods, a short circuit in the cables or connections between the sensing chamber and the electronics prevented any measurement (see Appendix A6 for further details). Note that in the balloon-borne configuration of the instrument, measurements have been performed at even colder temperatures than those experienced in AIDA.

#### 3.5. Reference instruments

#### 3.5.1. APicT and SP-APicT tunable diode laser spectrometers

Two fiber-coupled tunable diode laser hygrometers (APicT and SP-APicT) measure the water vapor concentration at the AIDA chamber based on its absorption at 1370 nm. Both instruments are designed to selectively detect interstitial water vapor inside clouds and to continuously determine absolute water vapor mixing ratio inside the AIDA chamber (Ebert et al., 2005; Skrotzki, 2012; Skrotzki et al., 2013; Fahey et al., 2014; Sarkozy et al., 2020, Lamb et al., 2023). While APicT is connected to White-

cell optics allowing for a variable optical path between typically 23–99 m (200 m optionally), SP-APicT has an optical path length of only about 4 m as passing once across the chamber diameter. For the in situ measurement of water vapor concentration and partial pressure, the time resolution is approximately 1 s and the accuracy is given at ±5 %. The total water content is retrieved by extractive sampling of AIDA gas via a heated stainless-steel line in which ice and water droplets evaporate. To this line, the frostpoint mirror hygrometer MBW-373LX (see below) is connected. From the difference of total water and water vapor measurements, the condensed or cloud water content within AIDA can be derived. The water partial pressure obtained by the TDL instruments can be converted into an ice saturation ratio or relative humidity using the water vapor saturation pressure with respect to the AIDA gas temperature (Murphy and Koop, 2005). The accuracy of the retrieval of the relative humidity is therefore not only determined by the accuracy of the water vapor pressure but also by the uncertainty of the gas temperature. The AIDA mean gas temperature is determined by averaging over a representative number of 24 calibrated thermocouples distributed over the chamber volume in horizontal and vertical chains.

# 3.5.2. MBW-373LX dewpoint mirror hygrometer

10

The MBW-373LX is a chilled-mirror frostpoint hygrometer from MBW Calibration AG, Switzerland (now Process-Insights Inc.) that is used regularly in the AIDA facility. The unit has a frost-point accuracy of ±0.1 K traceable to calibration standards. The MBW-373LX instrument is located outside the AIDA chamber. It is connected to the chamber via 10-mm stainless-steel tubes ranging 40 cm into the chamber volume. With a 3-way valve either an inlet located at level 2 or at level 3 of the AIDA chamber can be selected. Both sampling tubes are heated to 30 °C from the inlet to outside of the cold box. For this study only the direct connection to the instrument on level 2 of the AIDA chamber was used with a total volume of 0.4 litre. Typical flow rates through the instrument are 1.0 SLM at a total chamber pressure of 1000 hPa decreasing to about 0.3 SLM at a pressure of 100 hPa. For lower pressures, the smaller flow rate leads to a slower time response of the instrument. Lower water vapor concentrations also typically result in a slower time response of the frost point mirror in the order of several minutes, since it requires more time to build up a detectable frost layer. Reference measurements at a water vapor pressure corresponding to saturation at 213 K showed a time resolution of 30 s of the instrument.

#### 4. Experiments and data analysis

The AquaVIT-4 campaign was divided in two phases: an "open intercomparison", carried out between 29 March and 1 April 2022, and a "blind intercomparison", carried out between 4–7 April 2022. Each phase comprised four experimental days, with each day consisting of 7–9 hours of measurements in the AIDA chamber.

During the open intercomparison, the temperature, pressure and H<sub>2</sub>O content of the AIDA chamber were defined jointly by the instrument team members, and "quicklooks" (i.e., preliminary evaluations) of the data by all instruments could be freely

exchanged between the groups. This phase provided a valuable first intercomparison dataset, while allowing to optimize the operation of the AIDA chamber and the design of the experiments.

During the blind intercomparison, the conditions of the AIDA chamber were determined by an independent board of referees, not affiliated with any participating instrument team, and unknown to the participants. The referee board consisted of the following members: M. Fujiwara (Hokkaido University, Japan), K. Rosenlof (NOAA, USA), and O. Möhler (KIT, Germany).

| Date       | Type  | Abbrev. | $T_{ m AIDA}$ [K] | p <sub>AIDA</sub> [hPa] | H <sub>2</sub> O APicT [ppm] | $N_{SP}$ |
|------------|-------|---------|-------------------|-------------------------|------------------------------|----------|
| 2022-03-29 | Open  | OD1     | $226 \pm 2$       | 120–500                 | 30–200                       | 7        |
| 2022-03-30 | Open  | OD2     | $244 \pm 3$       | 100–600                 | 100-530                      | 8        |
| 2022-03-31 | Open  | OD3     | $209 \pm 3$       | 20–325                  | 3–20                         | 6        |
| 2022-04-01 | Open  | OD4     | $194 \pm 2$       | 70–180                  | 0.5–2                        | 5        |
| 2022-04-04 | Blind | BD1     | $220 \pm 3$       | 20–250                  | 1–80                         | 10       |
| 2022-04-05 | Blind | BD2     | $204 \pm 2$       | 20–150                  | 1.5–20                       | 10       |
| 2022-04-06 | Blind | BD3     | $190 \pm 1$       | 70–100                  | 0.5–3                        | 7        |
| 2022-04-07 | Blind | BD4     | $234 \pm 3$       | 50-400                  | 25-400                       | 13       |

Table 2. Simulated conditions in the AIDA chamber during each day of the intercomparison (open, blind), in terms of temperature ( $T_{\text{AIDA}}$ , mean  $\pm$  standard deviation), pressure ( $p_{\text{AIDA}}$ , min-max range), H<sub>2</sub>O mixing ratio (measured by APicT, min-max range) and number of different setpoints ( $N_{SP}$ , i.e., static intervals).

The conditions were defined by the referee board based on the typical ranges of  $H_2O$  mixing ratio in the UTLS available in the literature and on the operational constraints of the AIDA chamber. Then, each instrument team reported their final data independently by submitting them directly to the referee board, without having access either to the reference instrument measurements or to those of other instruments. Finally, the referee board created time-series plots of all the measurements (similar to Fig. 4) and disclosed the data.

An exception to the blind intercomparison protocol was made after the disclosure of the results to allow the revision of a minor portion of the blind intercomparison datasets by two participating instruments (Pico-Light H<sub>2</sub>O and SAWfPHY). The reasons for this exception are documented in detail in Appendix A5 and A6.

#### 4.1. Simulated conditions

10

During each day of measurements, the temperature of the AIDA chamber ( $T_{AIDA}$ ) was kept mostly constant, while pressure ( $p_{AIDA}$ ) and H<sub>2</sub>O mixing ratio were varied in steps, aiming to simulate the conditions of a given layer of the atmosphere (from the mid troposphere to the lower stratosphere). Changes in  $p_{AIDA}$  were achieved by the addition/removal of dry synthetic air

 $(22.5 \% O_2 \text{ in } N_2, H_2O < 3 \text{ ppm})$  from/to the chamber, while changes in  $H_2O$  mixing ratio were achieved by injecting a given amount of pure  $H_2O$  into the chamber, either before the start or during the course of each experiment. Overnight between the experiments, the chamber was evacuated to less than 0.01 hPa pressure.

The ranges of  $T_{AIDA}$ ,  $p_{AIDA}$  and  $H_2O$  mixing ratio (according to the APicT measurements) established during each day of both the open and blind intercomparison are summarized in Table 2. The first two days of the open intercomparison (OD1, OD2) focused on simulating mid to upper tropospheric conditions, with  $T_{AIDA}$  between 226–244 K and high  $H_2O$  content (~30–500 ppm). Then, the chamber was cooled down to simulate UTLS (209 K, OD3) and tropical tropopause (194 K, OD4) conditions, with their associated low  $H_2O$  content (< 20 ppm). Day 1 of the blind intercomparison (BD1) spans a wide range of pressures (20–250 hPa) and  $H_2O$  mixing ratios (1–80 ppm) at around 220 K, simulating both upper tropospheric and lower stratospheric conditions (separately). Then,  $T_{AIDA}$  was reduced to UTLS (204 K, 1.5–20 ppm  $H_2O$ , BD2) and tropical tropopause (190 K, 0.5–3 ppm  $H_2O$ , BD3) conditions. Finally, on BD4 the chamber was warmed up to 234 K to simulate mid and upper tropospheric conditions (with up to 400 ppm  $H_2O$ ).

10

Figure 2 displays the distributions of the mean  $H_2O$  mixing ratio (measured by APicT) against  $p_{AIDA}$  and  $T_{AIDA}$  (color-coded) of all the measured static intervals ("setpoints"), overlaid with two vertical profiles of  $H_2O$  mixing ratio in the atmosphere. The atmospheric profiles are measured by the CFH during recent field campaigns and correspond to moist, tropical summer conditions (dashed grey line) and dry, mid-latitude winter conditions (solid grey line). Open intercomparison and blind intercomparison setpoints are indicated by empty and filled dots, respectively. Note that the selection of the setpoint intervals is described in Section 4.3.

Figure 2 shows that the simulated conditions in the AIDA chamber during AquaVIT-4 cover well the expected range of variability of H<sub>2</sub>O in the UTLS, up to pressures of ~20 hPa (corresponding to approximately 28 km altitude), and for different latitudes and seasons. Furthermore, H<sub>2</sub>O mixing ratios that are below of those typically observed in the atmosphere (< 2 ppm) were also investigated, both in the open and blind intercomparison. These measurements represent extremely dry conditions that can be associated with tropical deep convection (typhoon) events, as for example during the overpass of the Pico-SDLA instrument over the typhoon Raï during the Strateole 2 campaign, with mixing ratios of 1.5–2 ppm H<sub>2</sub>O (Carbone et al., 2024).

Although rarely found in the atmosphere (e.g., Inai et al., 2012; Brunamonti et al., 2018; 2019), measuring mixing ratios below 2 ppm H<sub>2</sub>O is useful to assess the detection limits of the different instruments.

Figure 2. Scatter plot of pressure ( $p_{AIDA}$ ) vs. H<sub>2</sub>O mixing ratio (according to APicT measurements) and temperature ( $T_{AIDA}$ , color-coded) measured during all static intervals (i.e., setpoint) of the intercomparison, overlaid with two atmospheric profiles of UTLS H<sub>2</sub>O. The profiles correspond to tropical summer conditions (dashed grey line: Brunamonti et al., 2018) and mid-latitude winter conditions (solid grey line: Graf et al., 2021) and were smoothed by a  $\pm 1$  km moving average. Open intercomparison and blind intercomparison setpoints are indicated by open and filled circles, respectively.

#### 4.2. Measurements overview

15

Figures 3–4 show the time series of the  $H_2O$  mixing ratio measured by all instruments, along with pressure and temperature of the AIDA chamber, for each day of the open (Fig. 3) and blind (Fig. 4) intercomparison. As already mentioned, each day of experiment was performed at constant  $T_{AIDA}$ , while the  $H_2O$  mixing ratio varied depending on the amount of water added directly to the chamber at the beginning of each day, and on subsequent changes in pressure. Each combination of  $H_2O$  and  $p_{AIDA}$  was held constant for about 30–60 min, for a total of 5–13 atmospherically-relevant static conditions sets measured during each day. The transient temperature changes shown in Fig. 3–4 are the adiabatic response to rapid addition or removal of air from the chamber, while the walls and enclosure remain at constant temperature.

While generally the  $H_2O$  mixing ratio values were kept below the ice saturation level, ice saturated and even supersaturated conditions were also generated in the AIDA chamber at a few instances by forming ice clouds. Particularly, this can be seen in OD4 (Fig. 3f, time ~6–7 h), BD2 (Fig. 4b, time ~7 h) and BD3 (Fig. 4f, time ~9–10 h). During these conditions, due to the evaporation of the the ice crystals contained in the AIDA chamber during their transport inside the heated sampling tube, the extractive instruments (ALBATROSS and MBW373LX) measured the total (i.e., sum of gas and condensed phase)  $H_2O$  content of the chamber, whereas non-extractive instruments (all others) measured the gas-phase  $H_2O$  only. These measurements are particularly valuable since, assuming the ice cloud is dense enough to sustain a constant relative humidity over ice (RH<sub>Ice</sub>) of 100 %, they provide an intrinsic reference for the gas-phase (non-extractive) instruments, corresponding to the ice saturated  $H_2O$  mixing ratio, which can be calculated based on the  $p_{AIDA}$  and  $T_{AIDA}$  measurements. The in-cloud interval measured during BD2 is analyzed in detail in this respect in Section 5.1.

The measurements of the extractive and non-extractive instruments also differ systematically due to adsorption/desorption effects of H<sub>2</sub>O, particularly the desorption of H<sub>2</sub>O molecules from the inner walls of the sampling tube (especially at low pressure), increasing the H<sub>2</sub>O mixing ratio in the sample gas and causing a memory effect in the system. Since the rate of adsorption/desorption has a complex dependency on pressure, water content, and sampling time (e.g., note that the difference between extractive and non-extractive instruments is larger on OD1 compared to OD3, at similar pressure conditions: Fig. 4a, 4d), these effects are particularly difficult to quantify. For this reason, we decided to separately define reference values for extractive and non-extractive instruments (see Section 4.4).

Adsorption/desorption on the chamber walls can also affect the  $H_2O$  mixing ratio inside the AIDA chamber. An example of this can be seen in OD3 (Fig. 2e-2g, time ~3–9 h), when the  $H_2O$  mixing ratio in the chamber increases from ~3 ppm to ~20 ppm upon a decrease in  $p_{AIDA}$  from 325 to 20 hPa. Since no additional  $H_2O$  was added to the chamber during this interval, and the mixing ratio should remain constant as air is pumped from the chamber, this increase must be due to the desorption of  $H_2O$  molecules from the inner walls of the chamber (or other internal components).

Figure 3. Time series of  $H_2O$  mixing ratio measured by all instruments (panels a-b, e-f), namely MBW373LX (pink), APicT (dark grey), SP-APicT (light grey), Pico-Light (orange), SAWfPHY (blue), ALBATROSS (purple) and DLH (red), along with AIDA chamber pressure ( $p_{AIDA}$ ) and temperature ( $T_{AIDA}$ ) (panels c-d, g-h) for each day of the open intercomparison phase of the campaign. All data shown here are averaged to a uniform integration time of 2 s. The static intervals selected for the statistical comparison (i.e., setpoints) are identified by the grey shaded intervals (duration 10 min each). Note that, for consistency, the color coding applied here to the different instruments is used consistently in all the following figures. The initial phases especially of the experiments with lower water mixing ratios were intended to give the sampling lines time to dry. The solid black line in panels (a-b, e-f) shows the ice saturated  $H_2O$  mixing ratio ( $H_2O_{LeeSat}$ ) calculated from  $p_{AIDA}$  and  $T_{AIDA}$  (see Section 5.3).

Figure 4. Same as Figure 3 for the blind intercomparison phase of the campaign.

# 4.3. Statistical intercomparison

5

To quantify the differences between each participating instruments and their corresponding reference, a series of data intervals that provided nearly constant  $H_2O$  mixing ratio,  $p_{AIDA}$  and  $T_{AIDA}$  (hereafter "setpoints") were selected for each day. Setpoint intervals, with a duration of 10 min each, were defined individually based on the analysis of the measured time series, and placed typically before a change in pressure or the injection of  $H_2O$  in the AIDA chamber occurred. Segments with static conditions that are irrelevant for the atmosphere (e.g., low  $H_2O$  mixing ratios at high pressure, and viceversa) were excluded

from the statistical analysis. In total, 66 atmospherically relevant setpoints could be identified during the open (26) and blind (40) intercomparison, which are shown as grey shaded intervals in Fig. 3–4.

The data from all instruments is expressed as  $H_2O$  mixing ratio (or molar fraction) in units of  $\mu$ mol/mol (i.e, parts per million, ppm), and submitted with a raw time resolution of approximately ~0.3 s (Pico-Light), ~0.7 s (APicT, SP-APicT), ~0.9 s (AL-BATROSS), and ~1 s (DLH, SAWfPHY, MBW373LX). For the statistical comparison, the entire data set is first averaged to an integration time of 2 s, and the relative difference ( $\Delta H_2O$ ) between each instrument and reference are calculated for each time step. Note that the choice of reference instrument to be associated with each category of instrument (extractive vs. non-extractive sampling) and  $H_2O$  mixing ratio range is discussed in the next section. Then, for each setpoint, the mean relative difference and standard deviation are calculated as the mean and standard deviation of  $\Delta H_2O$  over the corresponding time period. Finally, to quantify the results in terms of  $H_2O$  mixing ratio range, the mean relative differences are further averaged over four ensembles of setpoints, defined by a given condition in the average  $H_2O$  mixing ratio of the reference instrument (e.g.,  $H_2O < 2$  ppm). For each range, the mean relative deviation ( $\mu$ ) and standard deviation ( $\sigma$ ) are calculated as the mean and standard deviation over all setpoints belonging to that range.

The results of the statistical comparison are discussed in Section 5.1. The equations used for the data analysis, as well as a list of all setpoints measured during the campaign, can be found in Supplementary material.

#### 4.4. Reference instruments

Before evaluating the results of the participating hygrometers, the differences between the three reference instruments are systematically investigated with the aim to define the most appropriate reference instrument for each participating instrument, depending on sampling method (extractive vs. in situ) and  $H_2O$  mixing ratio range.

In AquaVIT-1 (Fahey et al., 2014), in the absence of an accepted reference for establishing the absolute accuracy of the instruments, the reference value for each static interval was taken as the ensemble mean of a "core" subset of instruments, showing deviation within ±10 % from such reference value at all conditions. It is relevant to note here that the "core" subset of instruments of AquaVIT-1 included both internal and external instruments, and the MBW373LX was not included in this subset. This was due to the fact that the MBW373LX flow rate dropped below ~0.3 SLM whenever the AIDA pressures was lower than 100 hPa, leading to a very slow response time of the instrument and memory effect in the sampling line, and thus, a systematic deviation with respect to the non-extractive instruments (Fahey et al., 2014).

In our case, given the presence of a single external participant (ALBATROSS), which shares the same sampling line with the MBW373LX and has a comparable flow rate (see Section 3), we can argue that the two instruments are subject to the same artifacts, and use the MBW373LX data as a reference for ALBATROSS. Therefore, two reference values are defined for each setpoint: one "internal" reference ( $Ref_{Ext}$ ), applied to the in situ instruments (Pico-Light H<sub>2</sub>O, SAWfPHY, DLH), and one "external" reference ( $Ref_{Ext}$ ), applied to the extractive instrument (ALBATROSS).

Figure 5 shows the scatter plots of the mean relative difference between the three reference instruments, namely MBW373LX – APicT (panel a), MBW373LX – SP-APicT (b) and SP-APicT – APicT (c), as function of the measured  $H_2O$  mixing ratio, for all setpoints of the open (white dots) and blind (filled dots) intercomparison, and color-coded with  $p_{AIDA}$ .

Three main effects influence the behaviour of the reference instruments: (*i*) slower time response due to required drying of the sampling tube (surface effects), leading to a systematic high bias of the extractive measurements with respect to the internal ones (Fig. 5a-b); (*ii*) longer times required to build up a detectable frost layer on the MBW373LX mirror at low H<sub>2</sub>O mixing ratios; and (*iii*) additional internal water absorption affecting the retrievals of the short-path SP-APicT, caused by water vapor in the sealed laser compartment, leading to a systematic moist bias of SP-APicT with respect to APicT at low H<sub>2</sub>O mixing ratios (Fig. 5c, grey dots).

Figure 5a shows that larger differences between extractive and in situ measurements tend to be associated with lower pressures in the AIDA chamber (see range of 5–10 ppm H<sub>2</sub>O), which is consistent with the pressure-dependent desorption of H<sub>2</sub>O molecules in the sampling line (*i*). However, no direct correlation can be inferred, due to the involved memory effects. The slow response time of MBW373LX due to the build up of the frost layer (*ii*) is compensated by selecting static intervals with nearly constant H<sub>2</sub>O mixing ratios.

The moist bias due to the absorption of water vapor in the hermetically sealed laser compartment (*iii*) is a well known feature of some DFB-TDL systems, like SP-APicT, coupled by fibres to a measurement volume as at the AIDA chamber (Buchholz and Ebert, 2014). This additional absorption signal leads to a larger bias for the measurement with a shorter optical pathlength in the measurement volume. Therefore, SP-APicT should be used only for H<sub>2</sub>O mixing ratios >100 ppm (Skrotzki, 2012; this work) or be corrected for the additional absorption (e.g., by quantifying the additional absorption with measurements at lower total pressure in the AIDA chamber).

Consequently, the internal and external reference values for each static interval are defined according to Eq. 1–2. Specifically,  $Ref_{Int}$  is defined as equal to the APicT retrieval for all setpoints with mean H<sub>2</sub>O mixing ratio lower than 100 ppm, and equal to the SP-APicT retrieval for all setpoints with mean H<sub>2</sub>O mixing ratio higher than 100 ppm, while  $Ref_{Ext}$  is equal to the MBW373LX retrieval for all setpoints.

$$Ref_{Int} = \begin{cases} APicT, & if H_2O_{APict} < 100 \text{ ppm} \\ SP - APicT, & if H_2O_{APict} > 100 \text{ ppm} \end{cases}$$

$$Ref_{Ext} = MBW373LX$$

The uncertainty of the reference measurements is defined by the uncertainty of the individual reference instruments, given in Table 1. A comparison of the internal reference measurements with the ice saturated  $H_2O$  mixing ratio during experimental phases with dense ice clouds is discussed in Section 5.2 and demonstrates the quality of the reference data.

Figure 5. Scatter plots of the mean relative difference between the three reference instruments, namely MBW373LX – APicT (panel a), MBW373LX – SP-APicT (b) and SP-APicT – APicT (c), as function of the measured  $H_2O$  mixing ratio, calculated for all setpoints of the open (open circles) and blind (filled circles) intercomparison, and color-coded with AIDA pressure ( $p_{AIDA}$ ). The setpoints associated with  $H_2O$  mixing ratios lower than 20 ppm have been shaded in the two lower panels because of the limitations of the SP-APicT instrument at these low values (see Section 4.4).

# 5. Results

10

Here we analyze the performance of the instruments in terms of accuracy with respect to the reference measurements (Section 5.1) as well as precision (Section 5.2). A detailed analysis of an in-cloud measurement interval at ice saturation, providing an independent internal reference, is presented in Section 5.3.

# 5 5.1. Statistical intercomparison Accuracy

The results of the statistical comparison are presented in Figure 6 and summarized in Tables 3–4. Figure 6 shows the scatter plots of the mean relative difference ( $\mu$ ) of each instrument with respect to its reference dataset, namely ALBATROSS –  $Ref_{Ext}$  (panel a), Pico-Light H<sub>2</sub>O–  $Ref_{Int}$  (b), SAWfPHY –  $Ref_{Int}$  (b) and DLH –  $Ref_{Int}$  (d), calculated for all setpoints of both the open and blind intercomparison. Vertical error bars show the  $\pm 1~\sigma$  range of standard deviation of the difference (calculated according to Eq. 3), while horizontal error bars the standard deviation range of the reference H<sub>2</sub>O mixing ratio. Table 3 shows the mean values of  $\mu \pm \sigma$  for each instrument and setpoint of the blind intercomparison, calculated in five ranges of H<sub>2</sub>O mixing ratio, namely H<sub>2</sub>O<sub>Ref</sub> < 2 ppm, H<sub>2</sub>O<sub>Ref</sub> = 2–10 ppm, H<sub>2</sub>O<sub>Ref</sub> = 10–100 ppm, H<sub>2</sub>O<sub>Ref</sub> > 100 ppm, as well as all data together, along with the number of setpoints measured by each instrument in each range ( $N_{SP}$ ). Note that the range of each setpoint is assigned for each participating instrument depending on the mean H<sub>2</sub>O mixing ratio of the reference instruments associated with that participant (i.e.,  $Ref_{Ext}$  for ALBATROSS and  $Ref_{Int}$  for all other instruments).

H<sub>2</sub>O > 100 ppm. At higher H<sub>2</sub>O mixing ratios, the agreement between participants and reference instruments is generally very good. Most instruments (ALBATROSS, DLH, Pico-Light H<sub>2</sub>O) achieve a mean deviation smaller than  $\pm 4$  % with respect to the reference, with standard deviations of 1–2 % (see Table 3), and all measured setpoints within  $\pm 8$  %. The only exception is SAWfPHY, showing larger discrepancies (> 10 %) for three of the measured setpoints in this range, and a mean deviation of  $-7 \pm 19$  % (Table 3). This is due to the fact that during the second half of BD4, the AIDA temperature setpoint was higher than -40 °C (see Fig. 4h). Therefore, uncertainties on the phase condensate on SAWfPHY's sensor caused a biased estimation of the H<sub>2</sub>O partial pressure (see more details in Appendix A6).

 $H_2O = 10-100$  ppm. In this range, all instruments obtain a good agreement with the reference measurements. ALBATROSS and DLH achieve a mean accuracy of  $0 \pm 4$  % and  $0 \pm 2$  %, respectively, and all setpoints within a deviation of  $\pm 9$  % (ALBATROSS) and  $\pm 4$  % (DLH) with respect to their reference. For Pico-Light  $H_2O$ , the mean accuracy is  $-3 \pm 9$  % with all setpoints within  $\pm 18$  %, while SAWfPHY achieves a mean accuracy of  $-6 \pm 12$  % and deviations smaller than  $\pm 20$  % for all except one setpoint, which is again due to the dry bias observed in BD4.

 $H_2O = 2-10$  ppm. For  $H_2O$  mixing ratios between 2-10 ppm, the mean discrepancy of all instruments stays within  $\pm 8$  %, whereas the relative standard deviations tend to increase due to the reduced  $H_2O$  content. ALBATROSS and DLH tend to show a slight overestimation of the reference  $H_2O$  mixing ratios, with mean accuracies of respectively  $8 \pm 12$  % and  $5 \pm 5$  % over all setpoints of the blind intercomparison. Pico-Light  $H_2O$  and SAWfPHY show a slight underestimation of the reference

measurements, with mean deviations of respectively  $-8 \pm 5$  % and  $-2 \pm 3$  % (Table 3). The ALBATROSS standard deviation is strongly influenced by a single setpoint with a deviation ~40 % measured on BD1 at  $p_{AIDA}$  ~20 hPa (see Fig. 4a). For SAWfPHY, only less than half (6 out of 13) of the setpoints in this range were measured, due to the connection issues occurred on BD3 (see Section 3.2).

Figure 7 shows a detail of the time series of two selected measuring intervals in the range 2–10 ppm H<sub>2</sub>O, namely OD3 (panel a) and OD2 (b), along with the relative difference between each intstrument and its corresponding reference (c-d). One noteworthy feature shown here is the small-scale fluctuations in H<sub>2</sub>O (peak-to-peak amplitude ~0.12 ppm) observed in the extractive instruments ALBATROSS and MBW373LX. These fluctuations are a measurement artifact due to the heating controller of the sampling line, shared by both instruments, which modulates the temperature (hence the H<sub>2</sub>O mixing ratio, via temperature-induced adsorption/desorption effects on the inner walls of in-the sampling line) with a period of about 3.5 min. This artifact is not corrected in the analyzed data statistical comparison, hence the fluctuations appear in the difference between the two instruments as well (Fig. 7c-d). The slight phase offset between the two retrievals is likely due to the different flow rate/response time of the two instruments. The fluctuations are instead corrected by a sinusoidal fit for the precision analysis (see next section). Net of these fluctuations, the measurement precision of the two instruments is consistent with the values reported in Table 1.

The Pico-Light  $H_2O$  measurements depict an increased dispersion when the pressure increases with a decreasing mixing ratio. In Fig. 7a, the setpoint at ~4.2 ppm  $H_2O$ , where the dispersion is the largest, occurs at  $p_{AIDA}$  ~150 hPa, while the setpoint at ~5.3 ppm  $H_2O$  occurs for  $p_{AIDA}$  ~100 hPa (see Fig. 3g). Spectroscopic techniques are sensitive to the environmental thermodynamic conditions, as the shape of the the absorption line which is used to retrieve the mixing ratio depends on temperature and pressure. The measurement uncertainty is then also related to ambient conditions. In this case, the absorption depth is about 30 % larger at ~100 hPa compared to ~150 hPa. Hence, the instrument uncertainty is then better at lower pressures, and this leads to smaller variability in the retrievals.

 $H_2O 

Figure 6. Scatter plots of the mean relative difference ( $\Delta H_2O$ ) of each instrument with respect to its reference dataset, namely AL-BATROSS –  $Ref_{Ext}$  (panel a), Pico-Light –  $Ref_{Int}$  (b), SAWfPHY –  $Ref_{Int}$  (b) and DLH –  $Ref_{Int}$  (d), calculated for all setpoints of the open (open circles) and blind (filled circles) intercomparison. Vertical error bars show the standard deviation range of the difference within each setpoint, while horizontal error bars show the standard deviation range of the reference H<sub>2</sub>O mixing ratio.

5

|                        | ALBATROSS        |          | DLH              |          | Pico-Light       |          | SAWfPHY          |          |
|------------------------|------------------|----------|------------------|----------|------------------|----------|------------------|----------|
| H <sub>2</sub> O range | $\mu \pm \sigma$ | $N_{SP}$ |
| < 2 ppm                | 6 ± 8 %          | 5/5      | 6 ± 9 %          | 6/6      | 54 ± 55 %        | 5/6      | $-3 \pm 2 \%$    | 2/6      |
| 2-10 ppm               | $8\pm12~\%$      | 13/13    | 5 ± 5 %          | 13/13    | $-8 \pm 5 \%$    | 13/13    | $-2 \pm 3 \%$    | 6/13     |
| 10-100 ppm             | $0\pm4$ %        | 13/13    | 0 ± 2 %          | 12/12    | $-3 \pm 9 \%$    | 11/12    | -6 ± 12 %        | 11/12    |
| > 100 ppm              | $-3 \pm 1$ %     | 9/9      | 0 ± 1 %          | 9/9      | -4 ± 2 %         | 9/9      | $-7 \pm 19 \%$   | 8/9      |
| All setpoints          | $3\pm9$ %        | 40/40    | 3 ± 5 %          | 40/40    | 3 ± 28 %         | 38/40    | -5 ± 13 %        | 27/40    |

Table 3. Summary of mean deviations and their standard deviation ( $\mu \pm \sigma$ ) for each instrument during the blind intercomparison, calculated in five ranges of H<sub>2</sub>O mixing ratio, along with the number of setpoints measured by each instrument in each range ( $N_{SP}$ ). Note that the range of each setpoint is assigned for each participant instrument based on the mean H<sub>2</sub>O mixing ratio of the reference instruments associated with that participant (i.e.,  $Ref_{Ext}$  for ALBATROSS,  $Ref_{Int}$  for all other instruments).

Figure 7. Detail of the time series of two selected measuring intervals in the range 2–10 ppm H<sub>2</sub>O, namely from the OD2 (panels a, c) and BD2 (panels b, d). Panels (a-b): H<sub>2</sub>O mixing ratio measured by all instruments (same color coding as in Fig. 3-4) at integration time 2 s. Panels (c-d): relative difference (in percent units) between each instrument and its corresponding reference. In all panels, setpoint intervals are highlighted by the grey shaded areas.

5

15

Figure 8 shows a scatter plot of the reference  $H_2O$  mixing ratio and the mean chamber-pressure level ( $p_{AIDA}$ ) of each setpoint, color-coded with the absolute value of the mean relative difference ( $|\Delta H_2O|$ ) of each instrument with respect to its reference, and overlaid with two vertical profiles of  $H_2O$  mixing ratio in the atmosphere (as in Fig. 2). This highlights that for extractive instruments (ALBATROSS), the largest relative deviations occur at low  $H_2O$  mixing ratios and low pressures, where surface effects are enhanced, and hence an accurate determination of the  $H_2O$  mixing ratio is more difficult. Conversely, for in situ instruments (all others), the results are mainly affected by the  $H_2O$  mixing ratio range alone. Particularly, Pico-Light  $H_2O$  shows the largest discrepancies (< 20 %) at  $H_2O < 2$  ppm, SAWfPHY at  $H_2O > 100$  ppm (due to the dry bias on BD4), while DLH shows moderate discrepancies (

Figure 8. Scatter plot of  $p_{AIDA}$  against the reference  $H_2O$  mixing ratio ( $Ref_{Ext}$  for ALBATROSS,  $Ref_{Int}$  for all other instruments), color-coded with the absolute value of the mean relative difference ( $|\mu|$ ) of each instrument with respect to its reference dataset, for all setpoints of the intercomparison, overlaid with two atmospheric profiles of  $H_2O$  mixing ratio (same as in Fig. 2). Open intercomparison setpoints are show as empty dots, while blind intercomparison ones as filled dots.

#### 5.2. Precision

15

The precision of the instruments is evaluated by analyzing the timeseries of the setpoint measured at ~6 ppm H<sub>2</sub>O and ~100 hPa pressure during OD3 (Fig. 7a, time ~6.4 h), representing average tropopause conditions. Figure 9 (left column) shows a zoom-in of the timeseries of H<sub>2</sub>O mixing ratio at 2 s resolution measured during this period by all instruments (excluding SP-APicT, since the H<sub>2</sub>O level considered here is well below the range of applicability of this instrument).

Following Fahey et al. (2014), a linear fit is applied to all timeseries (black dashed lines in Fig. 9, left column) to define the time evolution of the H<sub>2</sub>O mixing ratio. This allows to distinguish instrumental variability from small changes in H<sub>2</sub>O occurring inside the AIDA chamber and/or sampling lines during this interval. For the extractive instruments (MBW373LX and ALBA-TROSS), an additional sinusoidal fit is applied (solid black line in Fig. 9a, 9m) to account for the small-scale fluctuations in H<sub>2</sub>O mixing ratio associated with the heating controller of the sampling line (discussed in the previous section). For SAW-fPHY, due to a data gap occurred during this setpoint, two individual linear fits are applied to describe the two intervals of the

timeseries (see Fig. 9j). The exact form of the fitting equations and the corresponding parameters can be found in Supplementary material (Section S3).

The resulting detrended timeseries (i.e., measured data – fit) are shown in Figure 9 (middle column). The precision of each instrument is quantified by calculating the frequency of occurrence distributions (Fig 9, right column) and standard deviation at 2 s resolution ( $\sigma_{2s}$ ) of the detrended timeseries over the entire 10 min interval. The values of  $\sigma_{2s}$  obtained for each instrument are noted in the corresponding panels of Fig. 9 (right column), both as absolute and relative values (with respect to the mean of the considered instrument and setpoint). Note that the range of H<sub>2</sub>O values shown in Fig. 9 is ±0.1 ppm for all instruments except PicoLight H<sub>2</sub>O (Fig. 9g-i), for which a range of ±1 ppm is shown. The frequency of occurrence distributions are calculated in bins of 5 ppb for all instruments and 50 ppb for PicoLight H<sub>2</sub>O (Fig. 9i).

Overall, Fig. 9 shows that a higher precision is achieved on average by the instruments based on frostpoint hygrometry techniques (MBW373LX and SAWfPHY), with σ<sub>2s</sub> of respectively 6 ppb and 7 ppb, corresponding to approximately 0.1 % of the measured H<sub>2</sub>O mixing ratio. The laser spectrometers achieve precisions between 11 ppb (i.e., ~0.2 % of the measured signal) for APicT and ALBATROSS, 17 ppb (~0.3 %) for DLH, and 251 ppb (~4.7 %) for PicoLight H<sub>2</sub>O. All values obtained here are consistent with the uncertainty estimates given for each instrument in Table 1.

Figure 9. Precision analysis based on setpoint measured at  $\sim$ 6 ppm  $\rm H_2O$  and  $\sim$ 100 hPa pressure during OD3. Left column: timeseries of  $\rm H_2O$  mixing ratio measured by all instruments (colored dots) and corresponding linear fits (black dashed lines). Sinusoidal fits performed additionally for MBW373LX and ALBATROSS are shown as solid black lines (panels a, m). Middle column: detrended timeseries (i.e., difference between measured signals and the fitted curves). Right column: frequency of occurrence distributions, calculated in bins of 5 ppb (all instruments) and 50 ppb (PicoLight  $\rm H_2O$ , panel i). The standard deviation at 2 s resolution ( $\sigma_{2s}$ ) is noted for each instrument in the corresponding panel (both as absolute and relative values).

#### 5.32. In-cloud measurements

5

15

20

In this section, we analyze the in-cloud measurements performed during BD2. As already mentioned, these measurements are particularly valuable since, assuming the ice cloud is dense enough to sustain  $RH_{Ice} = 100$  % in the chamber (e.g., Fahey et al., 2014), they provide an intrinsic reference for the gas-phase (in situ) instruments, and hence allow to validate the quality of the internal reference instruments.

To this end, the ice saturated  $H_2O$  mixing ratio ( $H_2O_{IceSat}$ ) was calculated based on the  $p_{AIDA}$  and  $T_{AIDA}$  measurements using Eq. 3, assuming  $RH_{Ice} = 100$  % and the parameterization for saturation vapor pressure over ice ( $p_{Sat,Ice}$ ) from Murphy and Koop (2005). The uncertainty on  $H_2O_{IceSat}$  was also estimated based on the measurement uncertainties of  $p_{AIDA}$  and  $T_{AIDA}$ , which are 0.1 hPa in  $p_{AIDA}$  and 0.05 K in  $T_{AIDA}$ , plus a contribution up to 0.3 K due to temperature inhomogeneity in the chamber (Möhler et al., 2003). Assuming that the two uncertainty contributions in  $T_{AIDA}$  are uncorrelated, this corresponds for the conditions analyzed here ( $p_{AIDA} = 150$  hPa,  $T_{AIDA} = 204$  K) to an uncertainty in  $H_2O_{IceSat}$  of  $\pm 4.6$  % (i.e.,  $H_2O_{IceSat} = 21.2 \pm 1.0$  ppm).

(3) 
$$H_2O_{lceSat} = \frac{p_{Sat,Ice}(T_{AIDA})}{p_{AIDA}}$$

Figure 109a shows a time series of the H<sub>2</sub>O mixing ratio measured by all instruments during BD2, along with H<sub>2</sub>O<sub>IceSat</sub> calculated from  $p_{AIDA}$  and  $T_{AIDA}$  (black line) and its uncertainty (green shading), while Fig. 9b shows a zoom into the interval of ice cloud occurrence. The cloud is generated upon an addition of H<sub>2</sub>O into the chamber at constant  $p_{AIDA}$  and  $T_{AIDA}$  at time ~6.5 h. After the cloud formation, the gas-phase H<sub>2</sub>O mixing ratio measured by all in situ instruments is in good agreement with the ice saturated mixing ratio at around 21 ppm H<sub>2</sub>O, while the total H<sub>2</sub>O content measured by the extractive instruments increases up to around 50 ppm, indicating an ice water content of approximately 30 ppm H<sub>2</sub>O. Then, the total H<sub>2</sub>O decreases with time as the cloud-evaporates sublimates due to warming from the chamber walls (sedimentation may also contribute, depending on ice particle size), while gas-phase H<sub>2</sub>O remains stable until about the time of 7.1 h (see Fig. 109b). This indicates that the ice cloud was stable and dense enough to sustain RH<sub>Ice</sub> = 100 % in the chamber for approximately 40 min.

Figure 109c-d show the relative differences in gas-phase  $H_2O$  between all internal instruments and  $H_2O_{IceSat}$  (panel c), and the relative difference in total  $H_2O$  between ALBATROSS and MBW373LX (d) for the same time interval. All the in situ instrument measurements during the cloud occurrence interval fall within  $\pm 5$  % with respect to  $H_2O_{IceSat}$ . Particularly, APicT overestimates the ice saturated  $H_2O$  mixing ratio by  $\pm 5$  %, which is in line with at the upper edge of its nominal uncertainty range specified in Table 1. Pico-Light  $H_2O$  underestimates  $H_2O_{IceSat}$  by  $\pm 5$  %, while DLH shows an average deviation of approximately  $\pm 2.5$  % (i.e., within the  $\pm 4.6$  % uncertainty on  $H_2O_{IceSat}$  based on  $P_{AIDA}$  and  $T_{AIDA}$ ), and SAWfPHY varies between  $\pm 5$  % and  $\pm 2.5$  % (with a measurement gap in the first  $\pm 2.0$  min after cloud formation). The relative difference in total  $H_2O$  between ALBATROSS and MBW373LX ranges between  $\pm 4.0$  % and  $\pm 2.0$  % (Fig. 109d).

Figure 109. In-cloud measurements analysis. Panel (a): time series of H<sub>2</sub>O mixing ratio measured by all instruments during BD2 (same as Fig. 4b), along with the ice saturated H<sub>2</sub>O mixing ratio (H<sub>2</sub>O<sub>IceSat</sub>) calculated from *p*<sub>AIDA</sub> and *T*<sub>AIDA</sub> (black line) and its uncertainty (green shading). Panel (b): zoom into the ice cloud occurrence interval. Panel (c) relative differences (in percent) in gasphase H<sub>2</sub>O between all in situ instruments (APicT, SP-APicT, Pico-Light H<sub>2</sub>O, SAWfPHY, DLH) and H<sub>2</sub>O<sub>IceSat</sub>. Panel (d): relative difference in total H<sub>2</sub>O between the external instruments ALBATROSS and MBW373LX. In all panels, the blue shaded area highlights the interval of ice cloud occurrence (relative time 6.5–7.1 h).

#### 6. Conclusions

10

The AquaVIT-4 campaign continues the valuable efforts of the AquaVIT intercomparison series to assess the performance of atmospheric hygrometers for water vapor measurements in the upper atmosphere. This intercomparison involved four airborne hygrometers, deployed on aircraft or stratospheric balloon platforms and using either laser absorption spectroscopy or frost-point hygrometry techniques, which were compared with three in-house reference instruments at the AIDA chamber. This campaign provided an excellent opportunity to test new techniques, such as the Surface-Acoustic Wave frostpoint hygrometer (SAWfPHY) and the QCL-based ALBATROSS spectrometer, alongside established instruments, like the Pico-Light H<sub>2</sub>O (successor of the former Pico-SDLA) and the DLH spectrometer.

A strong focus was placed on simulating realistic UTLS conditions in the AIDA chamber (pressure 20–600 hPa, temperature 190–245 K,  $H_2O$  mixing ratio 0.5–530 ppm) and defining reference values for each static interval. Repeated measurements of the various experimental conditions highlight the need to distinguish between in situ and extractive methods, due to adsorpion/desorption effects of  $H_2O$  in the sampling line of the extractive instruments, and as such to define separate reference for them. Specifically, in situ instruments (DLH, Pico-Light  $H_2O$ , SAWfPHY) are compared with the APicT and SP-APicT reference spectrometers for setpoints the in the range of  $H_2O < 100$  ppm and  $H_2O > 100$  ppm, respectively, whereas extractive instruments (ALBATROSS) are compared with the MBW373LX dewpoint mirror hygrometer.

The statistical analysis shows that for  $H_2O > 100$  ppm, most instruments (ALBATROSS, Pico-Light  $H_2O$ , DLH) achieve mean deviation smaller than  $\pm 4$  % from the reference, while SAWfPHY shows larger discrepancies (between 10-20 %) for three individual setpoints. In the 10-100 ppm  $H_2O$  range, all instruments are in good agreement, with mean deviations within  $\pm 7$  % from the reference. This performance is also maintained in the range of 2-10 ppm  $H_2O$ , with mean deviations within  $\pm 8$  %, while the relative standard deviations tend to increase due to the reduced  $H_2O$  content. The largest differences were found in the range of  $H_2O < 2$  ppm, particularly for Pico-Light  $H_2O$ , which was not designed for such low mixing ratios. DLH, ALBATROSS and SAWfPHY achieve a good performance (within  $\pm 10$  %) also below 2 ppm  $H_2O$ .

10

For context, the subset of "core" instruments in AquaVIT-1 was found to agree to within ±10 % with the reference in the range 1–150 ppm H<sub>2</sub>O, and showed differences of –100 % to +150 % in the range H<sub>2</sub>O < 1 ppm (Fahey et al., 2014). While a direct comparison with these results is difficult due to the different reference value definitions, in general we observe slightly smaller mean deviations for the instruments participating in AquaVIT-4, both in the range of H<sub>2</sub>O > 10 ppm (±7 %) and 2–10 ppm H<sub>2</sub>O (±7 %), with respect to the AquaVIT-1 "core" instruments (±10 %). It is also remarkable that, unlike most of the AquaVIT-1 "core" instruments, the AquaVIT-4 instruments (DLH, ALBATROSS, SAWfPHY) achieve a good agreement (±10 %) at H<sub>2</sub>O < 2 ppm as well.

The measurement precision at 2 s resolution ( $\sigma_{2s}$ ) was evaluated for all instruments at average tropopause conditions (~6 ppm  $H_2O$  and 100 hPa pressure). This shows that a higher precision (6–7 ppb, corresponding to approximately 0.1 % of the measured  $H_2O$  mixing ratio) is achieved on average by the instruments based on frostpoint hygrometry techniques (MBW373LX and SAWfPHY). The laser spectrometers achieve precisions of 11 ppb (i.e., ~0.2 %: APicT and ALBATROSS), 17 ppb (~0.3 %: DLH), and ~250 ppb (~4.7 %: PicoLight  $H_2O$ ).

Of special interest were in-cloud conditions, i.e. intervals where ice saturation was reached and an ice cloud has been produced in the AIDA chamber. A detailed analysis of one of these intervals, where a dense ice cloud persisted for about 40 min, showed that all in situ (gas-phase) instruments fell within  $\pm 5$  % from the calculated ice saturation mixing ratio, which stands at the outer bound of the saturation mixing ratio uncertainty (~4.6 %, dominated by the uncertainty on measured air temperature). This demonstrates the reliability of the hygrometers in probing ice saturated conditions. Correspondingly, these instruments, if equipped with a highly accurate temperature sensor, can be used to study dehydrating processes across the tropopause. The  $\pm 5$  % agreement of APicT with respect to the calculated ice saturation H<sub>2</sub>O mixing ratio is consistent with the ice saturated

experiments performed in AquaVIT-1 (see Fig. A2 in Fahey et al., 2014), demonstrating that the performance of this instrument is preserved from the past AquaVIT-1 intercomparison.

In general, the observed agreement with the reference instruments was satisfactory for all the participants:

- For ALBATROSS, the deviations from MBW373LX obtained here fall within the expected range, although they are larger compared to a previous laboratory-based validation against SI-traceable reference gases (Brunamonti et al., 2023). This can be attributed to the uncertainty related to the residual water content (typically < 1 ppm) in the cell, originating from surface desorption effects along the sampling line. While these effects can be compensated by measuring a dry-air spectrum (Brunamonti et al., 2023), during AquaVIT-4 it was not possible to verify the zero level between the dynamic pressure conditions of the AIDA chamber, due to the very long equilibration times. Therefore, the "background" H₂O contribution was determined overnight, and parametrized as a function of cell pressure. This procedure, detailed in Appendix A3, can result in an uncertainty of up to ±0.2 ppm in the retrieved mixing ratios (i.e., ±20 % at 1 ppm H₂O), especially at low pressures (*p*<sub>AIDA</sub> ~20 hPa), which is about ten-fold larger than the analytical precision of ALBATROSS at these concentrations. It is important to note that these effects do not apply during flight conditions, where ALBATROSS operates in an open-path configuration (e.g., Graf et al., 2021), i.e., without any sampling line and with a gas flow of 3 orders of magnitude larger. Hence, the surface in contact with the gas is substantially smaller, and artifacts due to surface adsorption/desorption effects are negligible.
  - The performance of DLH is in line with expectations based on prior in-flight performance. As described in Section 3.2, the installation of DLH in the AIDA facility required the introduction of an additional window in the optical beam path, and with it the possibility of optical interference not present in the flight configuration, as well as an additional path between the instrument and the facility window which had to be kept sufficiently dry so as not to introduce an offset in the measured concentration. The size of the chamber limited the measurement absorption path-length to approximately 9.41 m, reducing the measurement sensitivity to between 52 % and 72 % of what it would be on the WB-57 aircraft. Despite these adaptations necessary to operate in the AIDA facility, DLH instrument performance appears to have met its stated accuracy and precision metrics during the AquaVIT-4 campaign.
- The Pico-Light H<sub>2</sub>O hygrometer is the successor of the former Pico-SDLA, which was operated during AquaVIT-1. In the range of H<sub>2</sub>O > 100 ppm, the Pico-SDLA agreed with the reference to within ±10 % on average (Fahey et al., 2014) while in AquaVIT-4, the agreement of Pico-Light H<sub>2</sub>O falls within ±4 %. In the range of 2–10 ppm H<sub>2</sub>O, during AquaVIT-1, Pico-SDLA had a tendency to overestimate mixing ratios, ranging from 20 % to more than 100 %. During AquaVIT-4, the agreement falls in the ±8 % range. The largest discrepancies observed during AquaVIT-1 were due to the simulated mixing ratios being dryer than normal at the given pressure levels. In such case, the absorption line depicts a flat broad shape which render difficult to separate the absorption signal to the spectrum baseline (zero absorption zones) and leads to large errors. During AquaVIT-4, the overestimations are reduced due to the fact that low mixing ratios have been simulated mainly at low *p*<sub>AIDA</sub> (10–150 hPa), where the line width permits a better identification of the zero absorption zones. For H<sub>2</sub>O < 2 ppm, due to the short path length (1 m), the sensitivity of the hygrometer

is not sufficient. The limit of quantification (10  $\sigma$ ) of the instrument is estimated to be of about 0.8 ppm H<sub>2</sub>O, close to the simulated conditions (see Appendix A5). The results of AquaVIT-4 are mostly in line with previous balloon-borne comparisons, i.e., within the  $\pm 10$  % range at UTLS conditions, though slightly larger than the claimed total uncertainty (see Ghysels et al., 2024: total uncertainty scaling from 4.4 % to 12% between 200 and 10 hPa).

SAWfPHY has been specifically designed to perform water vapor measurements onboard long-duration balloons flying in the tropical lower stratosphere. The instrument is therefore optimized for measurements in the sub-10 ppm H<sub>2</sub>O mixing ratios. In this range, SAWfPHY's measurements during both the open and blind periods of the AquaVIT-4 campaign achieved excellent accuracy and precision (< 3% down to 1.5 ppm), and are in line with the expected instrument performances. The specific configuration of SAWfPHY used in the AIDA chamber prevented the instrument to make measurements during some setpoints falling in this mixing ratio range. These limitations, however, do not apply to balloon flights (see Appendix A6), where observations have been recorded at temperatures comparable to the lowest ones experienced during the AquaVIT-4 campaign. SAWfPHY performances at mixing ratios higher than 10 ppm are generally in line with those in the sub-10 ppm range, except for a few setpoints that contribute to degrade the overall statistics. These lower-accuracy measurements directly result from the choices made in the instrument design and operations in the tropical lower stratosphere (see Appendix A6 for further details).</p>

Overall, the AquaVIT-4 campaign demonstrated the high accuracy and reliability of the four involved sensors for atmospheric monitoring and research applications, at a wide range of UTLS-relevant conditions. Here, it is also important to compare the results with the requirements on upper air  $H_2O$  measurement uncertainty defined by the GCOS Implementation Plan 2022 (GCOS, 2022). Based on the mean deviations found in this work (not considering vertical resolution), all instruments achieve at least the "threshold" level defined by GCOS (i.e.,  $\pm 10$  %) at all conditions (except for Pico-Light  $H_2O$  below 2 ppm), which qualifies the data as useful in terms of climate monitoring studies. Regarding the "breakthrough" and "goal" levels, we observe that the criteria that define these levels (2–5 %) are the same magnitude of the uncertainty of the reference instruments used here (see Table 1), which are state-of-the-art hygrometers in a well controlled laboratory setting as the AIDA chamber. Hence, although mean deviations smaller than 2–5 % are achieved by the instruments in individual  $H_2O$  ranges, an absolute assessment of these criteria is limited by the available reference methods.

#### Appendix. Data post-processing, calibration and instrument-specific issues

# A1. APicT, SP-APicT

30

The measurements with APicT and SP-APicT were done with 140 Hz and averaged to one second time resolution. The water absorption line profile of the 000-101, 110-211 transition at 7299.43 cm<sup>-1</sup> was fitted on-line with a Voigt profile based on line parameters from the HITRAN database (Rothman et al., 2012) and the actual temperature and pressure measured in the

AIDA chamber. No calibration was conducted and the absolute accuracy estimated from an error budget is better than 5 % and dominated by the line strength uncertainty (±3 %) (Fahey et al., 2014). Please note that the data used here were not post-processed and the "additional" water absorption in the laser was not subtracted. Therefore, the values at lowest water concentrations can show a high bias which is especially visible for SP-APicT data for mixing ratios below 20 ppm.

# 5 **A2. MBW373LX**

The data of the chilled-mirror hygrometer (MBW373LX, MBW calibration AG) was recorded on-line with a nominal time resolution of one second and an accuracy given by the manufacturer of  $\pm 0.1$  K frost point which corresponds to an accuracy in water partial pressure ranging from  $\pm 0.8$  % at 273 K to  $\pm 1.8$  % at 183 K (Murphy and Koop, 2005). The instrument was calibrated by the manufacturer on July 2020 in the range of 193 to 274 K (certificate: 8043MBW2020) documenting the traceability to national standards.

# A3. ALBATROSS

10

20

During AquaVIT-4, ALBATROSS was operated in a closed-path configuration, as described in Brunamonti et al. (2023). Such extractive configuration is only used for laboratory measurements, whereas for in-flight applications, the instrument is operated in an open-path configuration (Graf et al., 2021). In open-path configuration, the measurement takes place in situ at flow rates 3 orders of magnitude larger (~1500 vs 0.5 SLM), and the surface in contact with the gas is negligible. Therefore, the artifacts due to the adsorption/desorption of H<sub>2</sub>O molecules do not apply. The main reason for the extractive approach is due to the delicate electronics (DAQ, FPGA) and sensitive optical elements (IR-detector and QCL, requiring a temperature stability at mK level). While balloon flights typically last less than three hours, a period at which the implemented on-board heat-management can be fully granted, a continuous operation under low temperatures over two weeks, as was in the case of the AIDA chamber, is not supported.

A schematic of the sampling system used for ALBATROSS in AquaVIT-4 is shown in Figure A1. The segmented circular multipass cell (SC-MPC) is closed on both sides by electropolished stainless-steel lids, treated by a highly inert coating (SilcoNert® 2000, SilcoTek, USA) to minimize the adsorption of  $H_2O$  molecules on their surface. The sample gas is extracted from the AIDA chamber by a vacuum pump through a heated sampling line. The instrument is operated inside a custom-made PMMA plastic chamber to suppress any sudden temperature or humidity fluctuations. The pressure in the multipass cell ( $p_{Cell}$ ) is monitored by a heated capacitance manometer (AA02, MKS Instruments,USA), with an absolute accuracy of 0.12 %. The flow rate extracted from the AIDA chamber, and therefore  $p_{Cell}$ , varied during each experiment depending on the AIDA pressure. A mass-flow controller (MFC) upstream the SC-MPC was used to constrain the flow rate to a maximum of 0.5 SLM (corresponding to  $p_{AIDA} \sim 300$  hPa). Hence, all measurements at  $p_{AIDA} > 300$  hPa were performed at constant flow rate (0.5

SLM) and  $p_{Cell}$  (~45 hPa), while for  $p_{AIDA}$  

Figure A1. Schematics of the sampling system used for ALBATROSS in AquaVIT-4.

For the spectroscopic retrieval, the raw spectra are normalized to account for the laser intensity variation across the scanning range (baseline) and, in closed-path configuration, the "zero-air"  $H_2O$  contribution, i.e., the residual water content (typically < 1 ppm) in the cell due to surface desorption effects along the sampling line. The normalization can be done either by reconstructing the laser intensity baseline using a polynomial function (Graf et al.,2021), or by dividing each raw spectrum by the corresponding zero-air spectrum (i.e., the transmission through the multipass cell filled with  $N_2$  6.0) (Brunamonti et al., 2023). The second approach has the advantage that it accounts simultaneously for both the intensity baseline and the zero-air contributions. As this second contribution is pressure-dependent, it requires individual zero-air spectrum measurements for each investigated pressure level. During AquaVIT-4, due to the dynamic pressure conditions of the AIDA chamber, a direct access to such "zero-air" spectrum for each established pressure level was not possible, due to the long equilibration times. Therefore,

we adopted the polynomial-baseline approach, and subtracted the zero-air  $H_2O$  contribution after the spectroscopic retrieval, based on dry  $N_2$  (purity 6.0) measurements performed overnight between the experiments.

Figure A2 shows an overview of the nighttime measurement routine between BD2 and BD3. The zero-air spectra are acquired by measuring  $N_2$  at varying flow rate and pressure conditions. The resulting correction is then parametrized as a function of  $p_{Cell}$  and subtracted from the retrieved  $H_2O$  amount fractions. The MFC was programmed to scan the range 0–0.5 SLM in 32 steps of 10 min each, corresponding to 5–45 hPa in  $p_{Cell}$ , which covers the entire range of variability of  $p_{AIDA}$  during the daytime experiments. As expected, the zero-air  $H_2O$  mixing ratio (panel d) shows an exponential dependency on  $p_{Cell}$ , varying from about 10 ppm at  $p_{Cell}$  ~5 hPa (corresponding to  $p_{AIDA}$  ~15 hPa) and 0.7 ppm at  $p_{Cell}$  ~45 hPa ( $p_{AIDA} \ge 300$  hPa). The correction is calculated by fitting this behaviour using an exponential function. Zero-air measurements were repeated every night during the campaign, showing a slight tendency towards lower  $H_2O$  amount fractions with time.

The spectral fitting was performed using a quadratic speed-dependent Voigt profile (qSDVP) line shape model, with the optimized pressure broadening parameters ( $\Gamma_0$  and  $\Gamma_2$ ) derived by Brunamonti et al. (2023), and line strength parameter from the HITRAN2020 database (Gordon et al., 2022). This approach was found to substantially improve the accuracy of the instrument compared to the standard Voigt profile, in particular by removing pressure related artifacts (Brunamonti et al., 2023).

The zero-level correction described here represents the major source of uncertainty of the AquaVIT-4 measurements. Fig. 11e shows that this uncertainty is smaller than  $\pm 0.1$  ppm at most conditions, but increases to  $\pm 0.2$  ppm (i.e.,  $\pm 20$  % at 1 ppm H<sub>2</sub>O) at low pressures ( $p_{AIDA}$  ~20 hPa). Again, it should be noted that this procedure is not needed in normal open-path flight configuration.

Figure A2. Overview of the nighttime ALBATROSS zero-air ( $N_2$  6.0) measurements performed between BD2 and BD3. Panels (a-c): time series of the measured H<sub>2</sub>O mixing ratio, cell pressure ( $p_{Cell}$ ) and flow rate, respectively. Panel (d): H<sub>2</sub>O mixing ratio in the zero-air as function of  $p_{Cell}$  (blue markers) and exponential fit (red line), and the associated fit residuals (panel e).

#### A4. DLH

15

For each aircraft or ground facility, and each campaign, the DLH second harmonic absorption signal is converted to water vapor mixing ratio using quality-controlled temperature and pressure data and the measured optical path length. Using a WMS instrument model developed at NASA Langley, the range of temperatures and pressures for a given campaign, plus a range of

water vapor mixing ratios appropriate to each temperature and pressure, a matrix of line-center WMS 2f signal vs.  $(p, T, H_2O)$  is generated. Such a matrix is created for each absorption line, and for each set of WMS parameters. From these matrices, a multidimensional curve fit is created of the form  $H_2O = f(p, T, Signal, Line, Operational Parameters)$ . Each data point during the AquaVIT\_4 time period was converted in this way. All data were reported, except for time periods when the laser was not actively measuring water vapor. During the non-blind operating time period, opportunities were taken to collect data meant to improve DLH signal conversion, and data recorded during those time periods were not archived. Despite this, DLH data were reported during all blind- and non-blind comparison periods.

#### A5. Pico-Light H<sub>2</sub>O

5

10

The Pico-Light H<sub>2</sub>O hygrometer was located 1 m away for the SAWfPHY hygrometer in the main vessel. The optical cell was located in the main chamber and the electronic box was located in the laboratory space. The electronic box has been optimized to accommodate short-duration (2–3 hours) flights under radiosounding balloons, rather than long-duration balloon flights (as in Strateole-2). In the configuration of radiosounding flights, the electronics box needs less thermal insulation, and its heat capacity is sufficient to keep its temperature above the minimum operating temperature during the entire flight. Using less thermal insulation contributes to the decrease in weight. In the case of AquaVIT-4, the instrument remains several days at low temperatures. In such configuration, the electronic box has been deported in the laboratory space to preclude damages to the electronics of the instrument. To reproduce real flight configuration of the instrument, an empty electronic box has replaced the real one in the main vessel in order to address potential contamination effect.

The instrument acquires one unitary spectrum every 30 ms. The dataset provided for the open and blind intercomparison are the retrievals from the unitary spectra, using Pico-Light H<sub>2</sub>O internal pressure and temperature measurements. The H<sub>2</sub>O measurements are taken at 1 s intervals. During that interval, 200 ms are devoted to record the elementary atmospheric spectrum (within this time frame, five spectra are recorded), which comprises 1024 data points. The remaining 800 ms are used to record the atmospheric pressure and temperature, the GPS data, and the status of the instrument.

The mixing ratio is extracted from the absorption spectrum using a non-linear least-squares fitting algorithm applied to the full line shape, based on the Beer-Lambert law and in conjunction with in situ pressure and temperature measurements (Voigt profile). During a flight, two spectroscopic transitions are needed to probe both troposphere and stratosphere: the  $2_{02} \leftarrow 1_{01}$  (3801.41863 cm<sup>-1</sup>, "stratospheric line") and the  $4_{13} \leftarrow 4_{14}$  (3802.96561 cm<sup>-1</sup>, "tropospheric line") lines. During AquaVIT-4, setpoints simulated atmospheric conditions in the upper air. Therefore, only the stratospheric line was used. The line strength of the stratospheric line is about 10 times larger than the one of the tropospheric line.

The Pico-Light H<sub>2</sub>O dataset of part of BD4 (after relative time ~3 h) has been revised after the disclosure of the results. This is because the automatic data processing routine used to retrieve the mixing ratios from the recorded spectra was set to choose the tropospheric line instead of the stratospheric line, based on the measured air pressure (which is the approach used for flight data). However, as already mentioned, only the stratospheric line was used for AquaVIT-4. As a consequence, the retrieved

mixing ratio were 10 times larger than expected. Therefore, the data was then reprocessed using the appropriate spectral line, and the revised retrievals shared with the other participants and referees and used in this paper.

Figure A3 shows an example of a unitary spectrum recorded during BD3 for  $p_{AIDA} = 71.5$  hPa,  $T_{AIDA} = 190.5$  K and H<sub>2</sub>O mixing ratio = 0.81 ppm. This shows that the driest conditions simulated during AquaVIT-4 are close to the quantification limit of the instrument. The absorption depth of the stratospheric line is roughly 10 times larger than the noise level of the spectra. In such condition, the instrument uncertainty increases. The precision on mixing ratio is of about 0.28 ppm (no averaging), about 3 times smaller than the simulated mixing ratio. The total uncertainty for mixing ratio smaller than 1 ppm reaches 15–20 %.

Figure A3. Pico-Light  $H_2O$  unitary spectrum (10 ms integration time) of the "stratospheric line" at 3801.41863 cm<sup>-1</sup> during BD3 at extremely dry conditions ( $p_{AIDA} = 71.5$  hPa,  $T_{AIDA} = 190.5$  K, 0.81 ppm  $H_2O$ ) (panel a), and corresponding fit residuals (b).

#### 10 **A6. SAWfPHY**

Despite the excellent accuracy for most of the water vapor measurements performed by SAWfPHY, the instrument experienced a number of issues and did not reported measurements for some of the setpoints during both the AquaVIT-4 campaign weeks. We provide here more details on these events.

First, as already mentioned, the instrument experienced electrical issues between the electronics outside the vessel and the sensing chamber inside AIDA. These issues prevented any SAWfPHY measurements during the two coldest days (OD4 and BD3), leading to 12 missed setpoints in the range 0.5–3 ppm H<sub>2</sub>O. Yet, SAWfPHY reported measurements for H<sub>2</sub>O mixing ratios as low as 1.5 ppm during BD2, which are already rarely observed in the lower stratosphere. Another 4 setpoints at H<sub>2</sub>O mixing ratios < 5 ppm were missed by SAWfPHY in the beginning of BD1. These setpoints were associated with a difference between the AIDA and the frost-point temperatures larger than 30 K. Since SAWfPHY is a low-power instrument, the Peltier

device that cools the sensing surface is hardly able to provide such temperature difference. This has not prevented SAWfPHY to make measurements in the lower stratosphere during long-duration balloon flights as the difference between the frost-point and the air temperature is generally lower than 25 K there. In addition, part of this difference during balloon flights is achieved in a passive way by the radiator connected to the warm face of the Peltier element, which radiatively cools to space.

The SAWfPHY lowest precision measurements took place at H<sub>2</sub>O mixing ratios > 40 ppm. Those departures from the reference instrument are associated with SAWfPHY's control loop of the sensing device temperature, which has been optimized for lower stratospheric mixing ratios. The control loop of frost-point hygrometers is designed to compensate for the mass evolution of the condensate by adjusting the temperature of the sensing device. The temperature variations imposed by the control loop translate into a mass flux out of or into the condensate. For a given temperature variation, the more water available in the gas phase, the larger the mass flux. Hence, in order to measure H<sub>2</sub>O mixing ratios of a few ppm in a reasonable time, SAWfPHY's control loop has to be quite responsive, which can be detrimental at higher mixing ratios and lead to lower precision.

Last, a distinct issue took place on BD4, during which the AIDA temperature set point was higher than -40 °C. We discovered once the blind results have been revealed that SAWfPHY mixing ratios started to be strongly underestimated after a large injection of water in the AIDA vessel (about 5 h after the start of the experiment on that day). During this injection, the instrument control loop actually warmed the sensing surface of the instrument in order to maintain a constant mass of deposit on the sensing device. The sensing surface eventually reached temperatures higher than -40 °C, so that the ice deposit fully sublimated (or melted). A hypothesis for the subsequent underestimated measurements is that supercooled water droplets might have grown as a new condensate. Consequently, from that time on, we have corrected SAWPHY's measurements by using the saturation vapour pressure over supercooled water (Murphy and Koop, 2005), rather than over ice as is generally applied. This has significantly improved the comparison with the reference instrument, yet not perfectly. In any case, this issue is common to frost-point hygrometers, which have generally no way of knowing the phase of the condensate, and work best at temperatures lower than -40 °C or need to develop specific measurement strategies above -40 °C (e.g., Vömel et al., 2016).

#### Data availability

The data is available via the KIT open data repository (DOI will be added).

#### 25 **Supplement**

The supplement related to this article is available online at:

#### **Author contributions**

MG coordinated the AquaVIT-4 project and led the organization of the intercomparison campaign. SB merged the final datasets, performed the data analysis and generated all figures. HS controlled the AIDA chamber operation, supported the installation of all instruments and performed the reference measurements. MG, NA, RB, FF operated the Pico-Light H<sub>2</sub>O instrument and analyzed the data. SB, BT and LE operated the ALBATROSS instrument and analyzed the data. AH, PM, CC and JL

operated the SAWfPHY instrument and analyzed the data. GDi operated the DLH instrument and analyzed the data. MF, OM and KR acted as independent referees for the blind intercomparison. SB and MG wrote the paper with contributions from all authors.

# **Competing interests**

5 The authors declare that they have no conflict of interests.

## Acknowledgements

The authors gratefully acknowledge technical support by the AIDA team at KIT. The authors thank Julian Gisler (Empa) for his support in creating the graphics of Figures 1 and A1.

## **Financial support**

This work was supported by the European Commission under the Horizon 2020 – Research and Innovation Framework Programme, through the ATMO-ACCESS Integrating Activity under grant agreement No 101008004. The ALBATROSS participation was further supported within the framework of GAW/GCOS-CH (Swiss H<sub>2</sub>O-Hub project). The article processing charges for this open-access publication were covered by Empa (Switzerland).

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
