# Peer review of "The AquaVIT-4 intercomparison of atmospheric hygrometers"

_EGUsphere, 2025_

## Referee Comment (RC2)

**Review of: "The AquaVIT-4 intercomparison of atmospheric hygrometers" by Brunamonti et al.**

**Overall impression and rating**

The authors describe the laboratory comparison of four hygrometers in the Aida chamber in an excellent manner. The manuscript is of a very high standard, well structured, and easy to read. The figures are all clear and of excellent quality. The relevance to the community is also given because accurate water vapor measurements in UTLS are still very important for monitoring and process studies. I therefore recommend the manuscript for publication in AMT after a very few minor questions on my part have been answered.

**Specific comments/questions:**

- Page 2, lines 11-16: I think it should also be mentioned that negative trends have been found in stratospheric water (Hegglin et al. 2014), depending on the reference period used. This is clearly shown in Toa et al. 2023. I think it would be good to mention this as well, even though the paper is not about trends.

- Page, line 18: In the upper troposphere and even in the LMS higher mixing ratios above 10ppmv are observed. I would rather change the sentence to: "In the UTLS and in particular above the tropopause mixing typical mixing ratios of < 10ppmv are found."

- Page 3, lines 2-4: There are already alternatives for cooling frost point mirrors, such as dry ice or liquid nitrogen. The CFH for LN2 cooling can already be ordered from the manufacturer. I would therefore tone down the statement that there are already alternatives that still need to prove themselves in the future.

- Page 22, lines 6-8: Why should the temperature directly influence the water vapor mixing ratio ? Because of adsorption effects of the water vapor molecules on the tube wall ? You should add a short explanation here.

- Page 26, lines 5-7: How did you know the sampling efficiency of ice particles by the sampling line. Can you insure isokinetic sampling ? Otherwise you need to correct or it to determine the ice water content. Maybe it is worth mentioning this also in the text.

-

**Technical comments/suggestions:**

- Figure 3/4: I would suggest to include the saturation mixing ratio as additional line. This would help the reader identify which points in the time series are supersaturated or subsaturated.

**References**

- Hegglin, M., Plummer, D., Shepherd, T. et al. Vertical structure of stratospheric water vapour trends derived from merged satellite data. Nature Geosci 7, 768–776 (2014). https://doi.org/10.1038/ngeo2236

- Tao, M., Konopka, P., Wright, J.S. et al. Multi-decadal variability controls short-term stratospheric water vapor trends. Commun Earth Environ 4, 441 (2023). https://doi.org/10.1038/s43247-023-01094-9

---

## Author Comment (AC1)

Brunamonti et al. report a synthesis of results from the AquaVIT-4 project, providing a comparison of state-of-the-art atmospheric hygrometers for use in conditions found in the upper troposphere and lower stratosphere. Accuracy of hygrometers in these dry conditions is fundamental to assessment of ice microphysics, which controls the formation of cirrus and limits the transport of water vapor into the stratosphere. Substantial differences in the measurements reported in this region by various hygrometers has historically motivated a continued set of campaigns at AIDA and in the atmosphere, to assess the skill of research hygrometers.

This article is excellent. The experiment and analysis are well done, the figures are high quality and text is very well written. I essentially only have minor editorial comments for the authors to address, in addition to a couple of potential small changes that they could consider for the content and analysis.

**Authors:** We would like to thank the Referee for the constructive feedback that helped us to improve our manuscript. Below are the individual comments from the Referee (in black) and the replies from the Authors (in blue). Please note that page and line numbers given below refer to the revised manuscript without tracked-changes.

**Page7 Line 5:** Suggest to mention what kind of data acquisition mode is used for ALBATROSS, e.g. scanning with DC or 2f modulation, etc?

**Authors:** An additional paragraph was added to describe the method in more detail: *"ALBATROSS uses rapid spectral sweeping of the QCL by periodic modulation of the laser driving current. A highly energy-efficient strategy, referred to as "intermittent continuous-wave" (iCW) modulation (Fischer et al., 2014), is implemented, in which the driving current is applied in pulses, typically 200 μs long, followed by a short period of complete shutdown of the laser. The transmission data, consisting of $25 \times 10^3$ data points, are digitized by a 14-bit analogue–digital converter (ADC) at 125 MSs-1 and real-time processed by an FPGA (STEMlab 125-14, Red Pitaya). The signal-to-noise-ratio is further improved by averaging up to 3000 individual spectra in real time, leading to an effective measurement rate of 1 Hz."* (page 7, line 9).

**Page 8 Line 6:** Suggest instead of saying "25+ years" state the first year that it was operated.

**Authors:** Changed as requested ("*since 1994*").

**Page 9 lines 18 – 27:** Is this paragraph really needed? As I understand it, this is summarizing results that are already reported in Ghysels et al 2024. The large differences stated of "+/- 23.6%" and "about 30%" I assume are differences in the air that was measured by the instruments on those experiments and not due to accuracy problems with the instruments, but this isn't fully explained here. I would either provide more detail or remove that part.

**Authors:** Indeed, this paragraph summarizes the results of Ghysels et al. (2024). We agree with the Referee and hence removed the paragraph.

**Page 13 Line 15:** need to delete word "of"

**Authors:** Done.

**Figure 2:** Recommend using a discrete colorbar with ~5 degree intervals.

**Authors:** Implemented as suggested.

**Page 14 Line 7:** "instrument" -> "instruments"

**Authors:** Done.

**Page 26 Line 8:** "Evaporates" should be changed to "sublimates". However, I'm having trouble understanding that this is really the mechanism that results in a loss of ice. Is it rather that the ice crystals sediment to the bottom of the chamber? It doesn't make sense that the ice cloud would sublimate while it is supersaturated.

**Authors:** The cloud loses ice volume due to warming from the chamber walls, leading to sublimation. Sedimentation can contribute as well depending on the ice particle size. "Evaporates" was changed to "sublimates" and a short explanation was added: *"the total $H_2O$ decreases with time as the cloud  sublimates due to warming from the chamber walls (sedimentation may also contribute, depending on ice particle size)"* (page 28, line 19).

**Page 26:** Does the result in Figure 9 imply a positive bias in APicT and is it worth making a comment about how that might impact the comparisons shown previously where that hygrometer was the reference? Or is it rather the case that the uncertainty range shown for 100% RHi in Figure 9 is considered uniform such that the correct value could be anywhere in that +/-5% range with equal probability?

**Authors:** The deviation of +5 % between APicT and the ice saturated mixing ratio ($H_2O_{IceSat}$) is indeed at the upper edge of the APicT uncertainty range (±5 %), and might potentially indicate a positive bias of this instrument with respect to the true value. However, it should be considered that also $H_2O_{IceSat}$ has a comparable uncertainty (±4.6 %, green shading in former Fig. 9/now Fig. 10), due to the uncertainties on AIDA temperature and pressure. These uncertainties are derived from error budget calculations and account for both systematic and random effects. The measurement precision (i.e., random error component) is in both cases much smaller than the total uncertainty: ~0.1% for both APicT (see new Fig. 9, Section 5.2) and $H_2O_{IceSat}$ (estimated from the timeseries shown in former Fig. 9/now Fig. 10). Therefore, in both cases, the systematic error component dominates the uncertainty. This implies that, in the absence of an independent reference with a better uncertainty, the positive bias of APicT cannot be excluded, but also cannot be estimated based on this measurement.

The text was revised to point out that the observed discrepancy between APicT and $H_2O_{IceSat}$ is at the upper edge of the APicT uncertainty range (page 28, line 25).

One other comment I have is on the lack of substantial discussion or analysis about instrument precision. Precision is discussed a bit towards the end of the manuscript, but is not quantitatively summarized e.g. in Table 1 or assessed elsewhere. This paper would be a useful venue for comparison of the precision of the hygrometers as well as accuracy so I suggest that the authors consider addressing this somewhere. One suggestion if the authors do this is that some markers indicating the observed precision could be added in e.g. Figure 8, or a separate figure could be generated comparing the precision of the different instruments in this relevant P, H2O space.

**Authors:** We thank the Referee for this comment. We agree that a detailed assessment of the instrumental precision was lacking in the previous version, therefore we added a new dedicated Section ("5.2 Precision") and figure (Fig. 9) to the manuscript to address this aspect.

The corresponding text and figure can be found in the revised version of the manuscript (page 25, line 6 to page 27, line 7). In summary, the precision of the instruments was evaluated by a detailed analysis of the timeseries of a setpoint interval representing average tropopause conditions (~6 ppm $H_2O$ and ~100 hPa pressure). First, a linear fit was applied to all timeseries to define the time evolution of the $H_2O$ mixing ratio, allowing to distinguish instrumental variability from small changes in $H_2O$ occurring inside the AIDA chamber and/or sampling lines during the measurement interval. For the extractive instruments (MBW373LX and ALBATROSS), an additional sinusoidal fit was also applied to account for the small-scale fluctuations in $H_2O$ associated with the heating controller of the sampling line (discussed in Section 5.1). Then, the precision of each instrument was quantified by calculating the frequency of occurrence distributions and standard deviation at 2 s resolution ($\sigma_{2s}$) of the detrended timeseries over the entire (10 min) setpoint interval.

We believe this addition provides a very valuable insight to the paper, therefore we are grateful to the Referee for this suggestion. A short paragraph summarizing the results of the precision analysis was also added to the Conclusions section (page 30, lines 22-26).

Note that following this change, Section 5.1 was renamed (from "Statistical intercomparison" to "Accuracy"), and a short sentence was added to introduce the structure of Section 5.

---

## Author Comment (AC2)

**Review of: "The AquaVIT-4 intercomparison of atmospheric hygrometers" by Brunamonti et al.**

**Overall impression and rating**

The authors describe the laboratory comparison of four hygrometers in the Aida chamber in an excellent manner. The manuscript is of a very high standard, well structured, and easy to read. The figures are all clear and of excellent quality. The relevance to the community is also given because accurate water vapor measurements in UTLS are still very important for monitoring and process studies. I therefore recommend the manuscript for publication in AMT after a very few minor questions on my part have been answered.

**Authors:** We would like to thank the Referee for the constructive feedback that helped us to improve our manuscript. Below are the individual comments from the Referee (in black) and the replies from the Authors (in blue). Please note that page and line numbers given below refer to the revised manuscript without tracked-changes.

**Specific comments/questions:**

• **Page 2, lines 11-16:** I think it should also be mentioned that negative trends have been found in stratospheric water (Hegglin et al. 2014), depending on the reference period used. This is clearly shown in Toa et al. 2023. I think it would be good to mention this as well, even though the paper is not about trends.

**Authors:** We thank the Referee for this suggestion. The following sentence was added: "*The analysis of a composite of satellite observations showed negative trends in $H_2O$ in the lower and mid-stratosphere, and positive trends in the upper stratosphere, due to methane oxidation (Hegglin et al., 2014; Tao et al., 2023).*" (page 2, lines 17-18).

• **Page, line 18:** In the upper troposphere and even in the LMS higher mixing ratios above 10ppmv are observed. I would rather change the sentence to: "In the UTLS and in particular above the tropopause mixing typical mixing ratios of < 10ppmv are found."

**Authors:** Changed as suggested.

• **Page 3, lines 2-4:** There are already alternatives for cooling frost point mirrors, such as dry ice or liquid nitrogen. The CFH for LN2 cooling can already be ordered from the manufacturer. I would therefore tone down the statement that there are already alternatives that still need to prove themselves in the future.

**Authors:** We agree with this observation and rephrased the sentence as follows: "*Alternative cooling solutions, such as the use of liquid nitrogen or a mix of dry ice and alcohol, are currently being implemented and validated (e.g., Rolf et al., 2020; Dirksen, 2024; Poltera et al., 2025).*" (page 3, lines 4-6).

References:

Dirksen, R: R23 replacement (HP-2), GRUAN Implementation and Coordination Meeting (ICM-15), Bern, Switzerland, 11-15 March 2014, https://www.gruan.org/gruan/editor/documents/meetings/icm-15/pres/pres_0610b_Dirksen_R23.pdf (last access 5 July 2025), 2024.

Poltera, Y., Luo, B., Wienhold, F. G., and Peter, T.: Observations of water vapor in the UT/LS of unprecedented accuracy with non-equilibrium corrected frost point hygrometers, EGU General Assembly 2025, Vienna, Austria, 27 Apr–2 May 2025, EGU25-19811, https://doi.org/10.5194/egusphere-egu25-19811, 2025.

Rolf, C., Khordakova, D., and Vömel, H.: CFH cooling agent alternatives, GRUAN Implementation and Coordination Meeting (ICM-12), 16-20 November 2020, https://www.gruan.org/gruan/editor/documents/meetings/icm-12/pres/pres_302_Rolf_CFH_Cooling-Agent-Tests.pdf (last access 5 July 2025), 2020.

• **Page 22, lines 6-8:** Why should the temperature directly influence the water vapor mixing ratio? Because of adsorption effects of the water vapor molecules on the tube wall? You should add a short explanation here.

**Authors:** Indeed, temperature affects the $H_2O$ mixing ratio via temperature-driven adsorption/desorption effects on the inner walls of the sampling line. A short explanation was added to the manuscript: "*These fluctuations are a measurement artifact due to the heating controller of the sampling line, shared by both instruments, which modulates the temperature (hence the $H_2O$ mixing ratio, via temperature-induced adsorption/desorption effects on the inner walls of the sampling line)*" (page 22, lines 8-10).

 • **Page 26, lines 5-7:** How did you know the sampling efficiency of ice particles by the sampling line. Can you insure isokinetic sampling? Otherwise you need to correct or it to determine the ice water content. Maybe it is worth mentioning this also in the text.

**Authors:** Ice water content measurements from the same sampling system as used by MBW373LX and ALBATROSS during AquaVIT-4 were previously compared with in-situ measurements by FTIR, showing that a significant sampling loss only occurs for ice particle sizes larger than 7 μm (Haag et al., 2003). However, since a potential sampling loss would affect both instruments in the same way (as they share the same sampling line), this does not affect the comparison.

Reference:

Haag, W., Kärcher, B., Schaefers, S., Stetzer, O., Möhler, O., Schurath, U., Krämer, M., and Schiller, C.: Numerical simulations of homogeneous freezing processes in the aerosol chamber AIDA, Atmos. Chem. Phys., 3, 195–210, doi.org/10.5194/acp-3-195-2003, 2003.

**Technical comments/suggestions:**

• **Figure 3/4:** I would suggest to include the saturation mixing ratio as additional line. This would help the reader identify which points in the time series are supersaturated or subsaturated.

**Authors:** Done.

References

• Hegglin, M., Plummer, D., Shepherd, T. et al. Vertical structure of stratospheric water vapour trends derived from merged satellite data. Nature Geosci 7, 768–776 (2014). https://doi.org/10.1038/ngeo2236

• Tao, M., Konopka, P., Wright, J.S. et al. Multi-decadal variability controls shortterm stratospheric water vapor trends. Commun Earth Environ 4, 441 (2023). https://doi.org/10.1038/s43247-023-01094-9